# Learning Reward Machines from Partially Observed Policies

**Mohamad Louai Shehab**                                    *mlshehab@umich.edu*
*Department of Robotics*
*University of Michigan, Ann Arbor, USA*

**Antoine Aspeel**                                    *antoineaspeel@centralesupelec.fr*
*Universite Paris-Saclay, CNRS, Centrale-Supélec*
*Laboratoire des Signaux et Systèmes, Gif-sur-Yvette, France*

**Necmiye Ozay**                                    *necmiye@umich.edu*
*Department of Electrical Engineering and Computer Science*
*Department of Robotics*
*University of Michigan, Ann Arbor, USA*

**Reviewed on OpenReview:** *https://openreview.net/forum?id=7bbYYNvhTE*

## Abstract

Inverse reinforcement learning is the problem of inferring a reward function from an optimal policy or demonstrations by an expert. In this work, it is assumed that the reward is expressed as a reward machine whose transitions depend on atomic propositions associated with the state of a Markov Decision Process (MDP). Our goal is to identify the true reward machine using finite information. To this end, we first introduce the notion of a prefix tree policy which associates a distribution of actions to each state of the MDP and each attainable finite sequence of atomic propositions. Then, we characterize an equivalence class of reward machines that can be identified given the prefix tree policy. Finally, we propose a SAT-based algorithm that uses information extracted from the prefix tree policy to solve for a reward machine. It is proved that if the prefix tree policy is known up to a sufficient (but finite) depth, our algorithm recovers the exact reward machine up to the equivalence class. This sufficient depth is derived as a function of the number of MDP states and (an upper bound on) the number of states of the reward machine. These results are further extended to the case where we only have access to demonstrations from an optimal policy. Several examples, including discrete grid and block worlds, a continuous state-space robotic arm, and real data from experiments with mice, are used to demonstrate the effectiveness and generality of the approach.

## 1 Introduction

Several frameworks exist for solving complex multi-staged tasks, including hierarchical reinforcement learning (HRL) (Pateria et al., 2021), reward machines (RMs) (Icarte et al., 2018) and linear temporal logic (LTL) specifications (Chou et al., 2020; Vaezipoor et al., 2021). HRL leverages a decomposition of tasks into subtasks, enabling agents to focus on solving smaller, manageable problems before integrating solutions into a higher-level policy (Sutton et al., 1999). On the other hand, RM and its generalizations (Corazza et al., 2022) encode task-specific knowledge as finite-state machines, capturing temporal dependencies and logical constraints in a concise and interpretable manner, similar to LTL. This structure simplifies policy learning and improves efficiency, especially in environments with long horizons or sparse rewards.

As an extension to inverse reinforcement learning (IRL) (Ng and Russell, 2000), one could ask the question of learning RMs, which enables agents to autonomously extract structured representations of tasks, significantly enhancing their ability to solve complex, temporally extended problems. By learning reward

machines directly from demonstrations, agents can adapt to tasks without requiring manually specified task representations, making this approach scalable and practical for real-world applications, such as robotic manipulation and autonomous vehicle navigation (Camacho et al., 2019; Icarte et al., 2023; Xu et al., 2020; Baert et al., 2024; Camacho et al., 2021). This capability is especially valuable in environments where high-level task-relevant features (aka, atomic propositions) are observable, underscoring the importance of learning RMs in advancing autonomous decision-making systems. For instance, in a high-level indoor navigation or patrolling task, semantic room labels can act as such propositions. Beyond autonomy applications, IRL and RM learning can also be used to infer the agent's (e.g., humans') intentions to design incentives or better decision making environments (Nitschke et al., 2024). As will be demonstrated, such intent inference can also be used in neuroscience to analyze animal behavior and decision making.

Some of previous work on learning reward machines from data either assumes that the machine's states are observed (Araki et al., 2019) or the reward is observed (Xu et al., 2020; Icarte et al., 2023; Hu et al., 2024; Abate et al., 2023). In the latter case the problem becomes finding a reward machine consistent with the observed input-output traces. Other work (Hasanbeig et al., 2024; 2021; Furelos-Blanco et al., 2020) infers reward machines by combining automata synthesis with reinforcement learning and querying the environment for experiences. Others (Xu et al., 2021; Memarian et al., 2020) use the standard $L^*$ algorithm for automata learning (Angluin, 1987) to learn a consistent reward machine. This assumes access to an oracle that can answer membership and conjectures queries. There are also works that only use observations of atomic propositions (Camacho et al., 2021; Baert et al., 2024), similar to us; however, they are limited to single-stage goal reaching tasks, where the RM has a simple structure that is merely used to obtain dense rewards. In parallel, several works aim to infer an LTL specification from demonstrations satisfying and/or violating the specification (Neider and Gavran, 2018; Vazquez-Chanlatte and Seshia, 2020), requiring a potentially large, labeled data set. Since LTL learning problem is inherently ill-posed, several regularization techniques are used such as formula templates or concept classes.

To the best of the authors' knowledge, no prior work has formalized and solved the problem of learning reward machines from partially observed optimal policies directly without the need to observe the rewards or the machine's state. The two main challenges of this setting are 1) partial observability (the reward is not observed, only the atomic propositions are observed), 2) partial reachability (not all transitions of the reward machine are visited in a given environment). In this work, we address these challenges by first characterizing what can be learned in this setting (i.e., an equivalence class of reward machines) and then proposing a SAT-based algorithm, which provably learns a reward machine equivalent to the underlying true one. The key insight of our algorithm is to identify pairs of atomic proposition prefixes, namely *negative examples*, that lead to different nodes of the underlying reward machine from the observable optimal prefix-based policy, and encoding these examples as constraints in the SAT problem. We show that our method can be applied even when the optimal policy is accessible only through a finite set of optimal trajectories. To this end, we approximate the policy from the data and replace the SAT problem with a variant called weighted MAX-SAT that provides robustness to incorrectly labeled negative examples. We demonstrate the efficacy of our algorithm in diverse settings, including grid-based MDPs, a robotic control task, and a real-world dataset of mouse navigation.

**Notation:** Given a set $X$, we denote by $\Delta(X)$ and $|X|$ the set of all valid probability distributions on $X$ and the cardinality of $X$, respectively. $\mathbf{1}(X)$ denotes the indicator function of $X$. $X^*, X^\omega$ denote the set of all finite/infinite sequences of elements in $X$. For a sequence $\tau$ and non-negative integers $i, j$, $\tau_i$ denotes the $i^{th}$ element of $\tau$; $|\tau|$ denotes the length of $\tau$; $\tau_{end}$ denotes the last element of $\tau$ when $\tau$ is finite; $\tau_{i:j}$ denotes the subsequence starting with the $i^{th}$ element and ending with the $j^{th}$; and $\tau_{:i}$ denotes the subsequence ending with the $i^{th}$ element.

## 2 Preliminaries and Problem Statement

### 2.1 Markov Decision Processes and Reward Machines

A Markov Decision Process (MDP) is a tuple $\mathcal{M} = (\mathcal{S}, \mathcal{A}, \mathcal{P}, \mu_0, \gamma, r)$, where $\mathcal{S}$ is a finite set of states, $\mathcal{A}$ is a finite set of actions, $\mathcal{P} : \mathcal{S} \times \mathcal{A} \to \Delta(\mathcal{S})$ is the Markovian transition kernel, $\mu_0 \in \Delta(\mathcal{S})$ is the initial state

distribution, $\gamma \in [0, 1)$ is the discount factor and $r : \mathcal{S} \times \mathcal{A} \times \mathcal{S} \to \mathbb{R}$ is the reward function. The set of feasible state trajectories for an MDP $\mathcal{M}$, denoted by $\mathcal{T}_s(\mathcal{M})$, is defined as:

$$\mathcal{T}_s(\mathcal{M}) = \{(s_0, s_1, \ldots) \in \mathcal{S}^\omega \mid \exists (a_0, a_1, \ldots) \in \mathcal{A}^\omega : \mathcal{P}(s_{t+1} \mid s_t, a_t) > 0, \forall t\}.$$

When we want to refer to finite prefixes of $\mathcal{T}_s(\mathcal{M})$, we simply use $\mathcal{T}_s^{\mathrm{fin}}(\mathcal{M})$, and we omit $\mathcal{M}$ when it is clear from the context.

An MDP without the reward is referred to as an *MDP model*, and is denoted by $\mathcal{M}/r$. MDP models can be decorated with labels. We denote such labeled MDP models as $\mathcal{M}_L = (\mathcal{S}, \mathcal{A}, \mathcal{P}, \mu_0, \gamma, L, \mathrm{AP})$, where $L : \mathcal{S} \to \mathrm{AP}$ is a labeling function that assigns to each state an atomic proposition, representing high-level conditions satisfied at that state, from the set AP. A labeled MDP has a corresponding language $\mathcal{L}(\mathcal{M}_L) \subseteq (\mathrm{AP})^\omega$, with $\mathcal{L}(\mathcal{M}_L) \doteq \{\sigma \in (\mathrm{AP})^\omega \mid \sigma = L(\tau), \text{ where } \tau \in \mathcal{T}_s(\mathcal{M}_L)\}$, where we overload $L$ to take in sequences. We also define the prefixes of a language as:

$$\mathrm{Pref}(\mathcal{L}) = \{w \in (\mathrm{AP})^* \mid \exists x \in \mathcal{L}, \text{ s.t. } w \text{ is a prefix of } x\}.$$

The set of reachable states for a proposition sequence $\sigma$ is:

$$\mathrm{Reach}(\sigma) = \{s \in \mathcal{S} \mid \tau \in \mathcal{T}_s^{\mathrm{fin}} \text{ s.t. } L(\tau) = \sigma, \tau_{|\tau|} = s\}.$$

A Reward Machine (RM) is a tuple $\mathcal{R} = (\mathcal{U}, u_I, \mathrm{AP}, \delta_{\mathbf{u}}, \delta_{\mathbf{r}})$ which consists of a finite set of states $\mathcal{U}$, an initial state $u_I \in \mathcal{U}$, an input alphabet AP, a (deterministic) transition function $\delta_{\mathbf{u}} : \mathcal{U} \times \mathrm{AP} \to \mathcal{U}$, and an output function $\delta_{\mathbf{r}} : \mathcal{U} \times \mathrm{AP} \to \mathbb{R}$. To avoid ambiguity between MDP states and RM states, the latter will be referred to as *nodes*. The reward machine without the reward is denoted as $\mathcal{G} \triangleq \mathcal{R}/\delta_{\mathbf{r}}$, and we refer to it as a *reward machine model*. We extend the definition of the transition function to define $\delta_{\mathbf{u}}^* : \mathcal{U} \times (\mathrm{AP})^* \to \mathcal{U}$ as $\delta_{\mathbf{u}}^*(u, l_0, \cdots, l_k) = \delta_{\mathbf{u}}(\cdots (\delta_{\mathbf{u}}(\delta_{\mathbf{u}}(u, l_0), l_1), \cdots, l_k)$. Given a state $u \in \mathcal{U}$, we define *the paths of $u$* as the input words which can be used to reach $u$:

$$\mathrm{Paths}(u) = \{w \in (\mathrm{AP})^* \mid \delta_{\mathbf{u}}^*(u_I, w) = u\}.$$

We overload the operator Reach to include the set of MDP states reachable at $u$. It is given by:

$$\mathrm{Reach}(u) = \{s \in \mathrm{Reach}(\sigma) \mid \sigma \in \mathrm{Paths}(u)\}.$$

As a running example, we borrow the patrol task from (Icarte et al., 2018). Consider the room grid world shown in Figure 1a. It is a 4 by 4 grid where the agent can move in the four cardinal directions, with a small probability of slipping to neighboring cells. We color-code different cells to denote the proposition label associated with the corresponding cell. For example, all cells colored green have the high level proposition A. The agent is tasked to patrol the rooms in the order A $\to$ B $\to$ C $\to$ D. This is captured by the reward machine shown in Figure 1b. Assume the agent's state trajectory starts with $\tau = (\mathbf{a}_1, \mathbf{a}_2, \mathbf{a}_3, \mathbf{b}_4, \mathbf{c}_1)$. The proposition sequence associated with $\tau$ is $\sigma = AAABC$. The RM nodes traversed by following $\tau$ are $(u_0, u_1, u_1, u_1, u_2, u_3)$. Since $\mathbf{c}_1$ is the only state than can be reached with $\sigma$, we have that $\mathrm{Reach}(\sigma) = \{\mathbf{c}_1\}$. Similarly, $\sigma \in \mathrm{Paths}(u_3)$. Each transition in the reward machine gives a reward of zero, except the transition from $u_3$ to $u_0$, i.e., $\delta_{\mathbf{r}}(u, l) = 1$ if $(u, l) = (u_3, D)$, and zero otherwise.

## 2.2 Markov Decision Process with a Reward Machine

A Markov decision process with a reward machine (MDP-RM) is a tuple $\mathcal{R}_\mathcal{M} = (\mathcal{M}/r, \mathcal{R}, L)$ where $\mathcal{M}$ and $\mathcal{R}$ are defined as in Section 2.1, and $L$ is a labeling function $L : \mathcal{S} \to \mathrm{AP}$. An MDP-RM can be equivalently seen as a product MDP $\mathcal{M}_{\mathrm{Prod}} = (\mathcal{S}', \mathcal{A}', \mathcal{P}', \mu_0', \gamma', r')$ where $\mathcal{S}' = \mathcal{S} \times \mathcal{U}$, $\mathcal{A}' = \mathcal{A}$, $\mathcal{P}'(s', u'|s, u, a) = \mathcal{P}(s'|s, a)\mathbf{1}(u' = \delta_{\mathbf{u}}(u, L(s')))$, $\gamma' = \gamma$, $\mu_0' \in \Delta(\mathcal{S} \times \mathcal{U})$ with $\mu_0'(s, u) = \mu_0(s)\mathbf{1}(u = u_I)$ and $r'(s, u, a, s', u') = \delta_{\mathbf{r}}(u, L(s'))$. To make the notation compact, we denote the product state by $\bar{s} = (s, u)$. The product of an MDP model with a RM model is a product MDP model $\mathcal{G} \times \mathcal{M}_L = (\mathcal{S}', \mathcal{A}', \mathcal{T}', \mu_0', \gamma')$ defined similarly.

A *trajectory* of the product MDP $\mathcal{M}_{\mathrm{Prod}}$ is a sequence $(\bar{s}_\emptyset, a_\emptyset, \bar{s}_0, a_0, \bar{s}_1, a_1, \cdots)$, where $\bar{s}_\emptyset = (\emptyset, u_I)$ and $a_\emptyset = \emptyset$. An initial state $s_0$ is sampled from $\mu_0$. The introduction of $\bar{s}_\emptyset$ and $a_\emptyset$ at the start of the trajectory

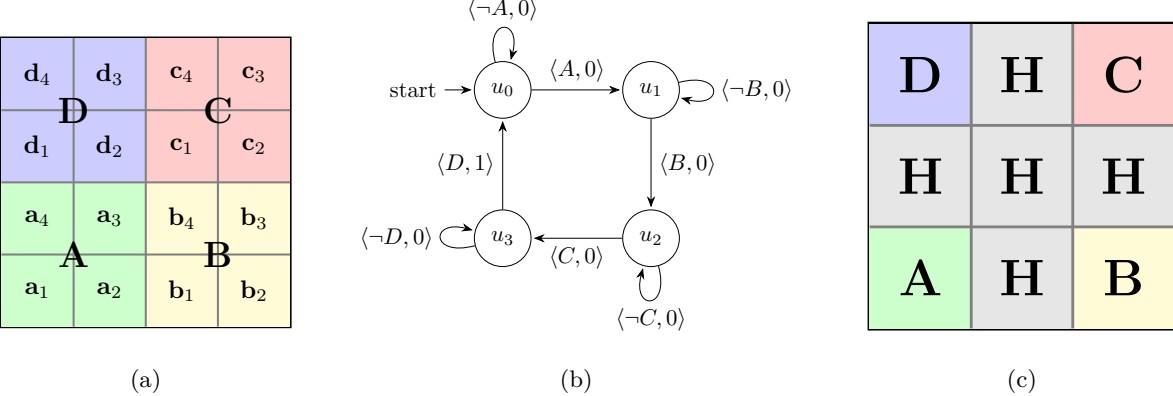

Figure 1: (a) The room grid world. (b) The patrol reward machine. (c) The room grid world with a hallway.

is to ensure that $s_0$ induces a transition in the reward machine. The reward machine thus transitions to $u_0 = \delta_{\mathbf{u}}(u_I, L(s_0))$. The agent then takes action $a_0$ and transitions to $s_1$. Similarly, the reward machine transitions to $u_1 = \delta_{\mathbf{u}}(u_0, L(s_1))$. The same procedure continues infinitely. We consider the product policy $\pi_{\mathrm{Prod}} : \mathrm{Dom}_{\mathrm{Prod}} \to \Delta(\mathcal{A})$ where $\mathrm{Dom}_{\mathrm{Prod}} \subseteq \mathcal{S} \times \mathcal{U}$ is the set of accessible $(s, u)$ pairs in the product MDP. This policy is a function that describes an agent's behavior by specifying an action distribution at each state. We consider the Maximum Entropy Reinforcement Learning (MaxEntRL) objective given by:

$$J_{\mathrm{MaxEnt}}(\pi; r') = \mathbb{E}_{\mu_0}^{\pi}\Big[\sum_{t=0}^{+\infty} \gamma^t \Big( r'(\bar{s}_t, a_t, \bar{s}_{t+1}) + \lambda \mathcal{H}(\pi(.|\bar{s}_t)) \Big)\Big], \tag{1}$$

where $\lambda > 0$ is a regularization parameter, and $\mathcal{H}(\pi(.|\bar{s})) = -\sum_{a \in \mathcal{A}} \pi(a|\bar{s}) \log(\pi(a|\bar{s}))$ is the entropy of the policy $\pi$. The expectation is with respect to the probability distribution $\mathbb{P}_{\mu_0}^{\pi}$, the induced distribution over infinite trajectories following $\pi$, $\mu_0$, and the Markovian transition kernel $\mathcal{P}'$ (Ziebart et al., 2008). The optimal policy $\pi_{\mathrm{Prod}}^*$, corresponding to a reward function $r'$, is the maximizer of (1), i.e.,

$$\pi_{\mathrm{Prod}}^* = \arg\max_{\pi} J_{\mathrm{MaxEnt}}(\pi; r'). \tag{2}$$

*Optimal* product MDP trajectories are trajectories of the product MDP generated using $\pi_{\mathrm{Prod}}^*$. We overload this definition to *optimal trajectories* of the MDP, which is generated from the optimal product MDP trajectories by simply removing the $u$ states. For the rest of the paper, *optimal trajectories* or *demonstrations* refer to the optimal trajectories of the MDP.

## 2.3 Prefix Tree Policy

Since the RM is unknown and the state of the RM is unobserved, we need a representation of the agent's policy that is independent of the RM state $u$. We accomplish this by defining a prefix tree policy (PTP) as the function that associates a distribution over the actions to each state and each finite sequence of atomic propositions that can be generated by the MDP. It is denoted as $\pi_{\mathrm{PTP}} : \mathrm{Dom}_{\mathrm{PTP}} \to \Delta(\mathcal{A})$, with $\mathrm{Dom}_{\mathrm{PTP}} = \{(s, \sigma) \mid \sigma \in \mathrm{Pref}(\mathcal{L}(\mathcal{M}_L)) \text{ and } s \in \mathrm{Reach}(\sigma)\}$. An important remark here is that the agent is acting according to a policy $\pi_{\mathrm{Prod}}(a|s, u)$, since the agent has access to $u$. The PTP in turn encodes the information of the agent's product policy in terms of the variables that we have access to, namely the MDP states only. The relation between the two policies is governed by:

$$\pi_{\mathrm{PTP}}(a|s, \sigma) = \pi_{\mathrm{Prod}}(a|s, \delta_{\mathbf{u}}^*(u_I, \sigma)), \tag{3}$$

where $\sigma \in \mathrm{Pref}(\mathcal{L}(\mathcal{M}_L))$. In particular, we say that the product policy $\pi_{\mathrm{Prod}}$ *induces* $\pi_{\mathrm{PTP}}$. We define the *depth-l restriction* of a PTP as its restriction to the set $\big(\cup_{j=1}^{l} \mathcal{S} \times (\mathrm{AP})^j\big) \cap \mathrm{Dom}_{\mathrm{PTP}}$. That is the policy associated to words $\sigma$ of length up to $l$. It is denoted by $\pi_{\mathrm{PTP}}^l$.

The induced PTP captures both what is observable about a product policy and what is reachable on the product. Therefore, we can only learn a reward machine up to whatever information is available in its induced PTP. We formalize this with the following definition.

**Definition 1.** *Two reward machines are **policy-equivalent** with respect to a labeled MDP model if the product policies obtained by solving problem (2) for each of the reward machines induce the same prefix tree policy defined as in (3). Among all the reward machines that are policy equivalent with respect to a labeled MDP, we define a **minimal** reward machine as one with the fewest number of nodes.*

Several equivalence relations among reward machines in the literature are special cases of policy equivalence. For instance, when learning finite state machines from observed rewards (Xu et al., 2020; Giantamidis et al., 2021), two reward machines are said to be input-output equivalent if they produce the same reward sequence for the same atomic proposition sequence. Such input-output equivalent reward machines are clearly policy-equivalent. Furthermore, for a trivial, i.e., one state, reward machine, our definition reduces to the policy-equivalence definition in the standard inverse reinforcement learning problem (Shehab et al., 2024; Cao et al., 2021).

### 2.4 Problem Statement

Consider a labeled MDP model $\mathcal{M}_L$ and a prefix tree policy $\pi_{\mathrm{PTP}}^{\mathrm{true}}$ induced by an optimal solution of problem (2). We are interested in the following two problems in this paper:

(P1) Does there always exist a depth-$l^*$ such that, given the labeled MDP model $\mathcal{M}_L$, a bound $u_{\max}$ on the number of nodes of the underlying reward machine, and the depth-$l^*$ restriction $\pi_{\mathrm{PTP}}^{\mathrm{true},l^*}$ of the true prefix tree policy, it is possible to learn a reward machine that is policy-equivalent to the underlying one?

(P2) If $l^*$ in problem (P1) exists, find a minimal reward machine that is policy-equivalent to the underlying one.

In what follows, we first provide an algorithm that takes the labeled MDP model $\mathcal{M}_L$, the depth-$l$ restriction $\pi_{\mathrm{PTP}}^{\mathrm{true},l}$ of the prefix tree policy $\pi_{\mathrm{PTP}}^{\mathrm{true}}$ for some arbitrary $l$, and the bound $u_{\max}$, and computes a reward machine that induces a prefix tree policy $\pi_{\mathrm{PTP}}^{\mathrm{learned}}$ with the same depth-$l$ restriction, i.e., $\pi_{\mathrm{PTP}}^{\mathrm{learned},l} = \pi_{\mathrm{PTP}}^{\mathrm{true},l}$. Then, we prove the existence of a sufficient depth-$l^*$ in (P1), for which this algorithm solves problem (P2). We provide an upper bound on $l^*$ in terms of the number of states in the MDP and the number of nodes in the RM.

## 3 Methodology

### 3.1 SAT Encoding

We encode the RM learning problem into a Boolean Satisfiability problem (SAT). SAT is the problem of determining whether there exists an assignment to variables of a given Boolean formula that makes it evaluate to true. While SAT is known to be NP-complete (Cook, 1971), there are several powerful off-the-shelf solvers capable of solving large practical instances (Biere et al., 2009). Specifically, we use SAT to encode a graph with $n \leq u_{\max}$ nodes and associate a Boolean variable with each edge in the graph. Each node has $|\mathrm{AP}|$ outgoing edges. We define the Boolean variables $\{b_{ikj} \mid 1 \leq i, j \leq n, 1 \leq k \leq |\mathrm{AP}|\}$ as:

$$b_{ikj} = \begin{cases} 1 & \text{if } i \xrightarrow{k} j, \\ 0 & \text{Otherwise}, \end{cases} \tag{4}$$

where we use the shorthand $i \xrightarrow{k} j$ to denote that proposition $k$ transitions node $i$ to node $j$, i.e., $\delta_{\mathbf{u}}(u_i, k) = u_j$. We can encode several properties of the RM into Boolean constraints. Without loss of generality, we set node 1 of the graph to be $u_I$. To make the derivation easier, we define for each atomic proposition $k$ an adjacency matrix $B_k$ with $(B_k)_{ij} = b_{ikj}$. The Boolean constraints we add are due to *determinism*, *full-specification*, *negative examples* and *non-stuttering* of the learned reward machine. We expand on each of them below.

**Determinism:** Due to the RM being a deterministic machine, each label can only transition to one node. The corresponding Boolean constraints are:

$$\forall i, k, j, \forall j' \neq j \quad b_{ikj} = 1 \implies b_{ikj'} = 0. \tag{5}$$

**Full Specification:** This constraint, also known as being input-enabled (Hungar et al., 2003), ensures that all labels generate valid transitions at all nodes. The corresponding Boolean constraints are:

$$\forall i, \forall k, \exists j \text{ such that } b_{ikj} = 1. \tag{6}$$

We can combine the conditions of Sections 3.1 and 3.1 into one condition on each $B_k$ enforcing that each row has exactly one entry with value 1.

**Negative Examples:** Our Boolean constraint here depends on the following result.

**Lemma 1.** *Let $\sigma, \sigma' \in (\mathrm{AP})^*$ be two finite label sequences. If $\pi_{\mathrm{PTP}}^{\mathrm{true}}(a|s, \sigma) \neq \pi_{\mathrm{PTP}}^{\mathrm{true}}(a|s, \sigma')$, then $\delta_{\mathbf{u}}^*(u_I, \sigma) \neq \delta_{\mathbf{u}}^*(u_I, \sigma')$.*

*Proof.* It follows from (3). $\qquad\square$

Based on this result, given the depth-$l$ restriction $\pi_{\mathrm{PTP}}^l$ of a PTP $\pi_{\mathrm{PTP}}$, we construct the set of negative examples as:

$$\mathcal{E}_l^- = \{\{\sigma, \sigma'\} \mid \pi_{\mathrm{PTP}}^l(a|s, \sigma) \neq \pi_{\mathrm{PTP}}^l(a|s, \sigma') \text{ for some } s, a\}. \tag{7}$$

Let $\sigma = k_1 k_2 \cdots k_l$ and $\sigma' = k_1' k_2' \cdots k_m'$ be two propositional prefixes that lead to different policies in the same state, therefore $\{\sigma, \sigma'\} \in \mathcal{E}_l^-$. We encode the condition given by Lemma 1 into Boolean constraints as:

$$(B_{k_l}^\intercal B_{k_{l-1}}^\intercal \cdots B_{k_1}^\intercal e_1) \bigwedge (B_{k_m'}^\intercal B_{k_{m-1}'}^\intercal \cdots B_{k_1'}^\intercal e_1) = \begin{bmatrix} 0 & 0 & \cdots & 0 \end{bmatrix}^\intercal, \tag{8}$$

where $\bigwedge$ is the element-wise **AND** operator and $e_1 \triangleq [1, 0, \cdots, 0]^\intercal$ indicates that the paths start from the initial node. Our algorithm adds the Boolean constraint given in (8) for each element of $\mathcal{E}^-$. The significance of encoding negative examples is that it eliminates the learning of trivial reward machines. In particular, our method never learns a trivial one-state reward machine with all self-transitions (Icarte et al., 2023) as long as there is at least one negative example in our prefix tree policy.

**Non-Stuttering:** A reward machine is said to be non-stuttering if when a proposition transitions into a node, that same proposition can not transition out of the node, i.e., for all $(u, a, u') \in \mathcal{U} \times \mathrm{AP} \times \mathcal{U}$: $\delta_{\mathbf{u}}(u, a) = u' \implies \delta_{\mathbf{u}}(u', a) = u'$. This is related to multi-stage tasks where the particular duration spent on a subtask (i.e., satisfying a given atomic proposition) is not important (Baier and Katoen, 2008). When the reward machine is *a priori* known to be non-stuttering, this extra condition can be included in the SAT problem. The main significance of this condition is trace-compression (Icarte et al., 2023), by which we can reduce the number of negative examples in $\mathcal{E}^-$ by only keeping the shortest negative examples among the equivalent ones. In this case, two negative examples are equivalent if between two pairs, the corresponding label sequences differ only by the same proposition repeated consecutively more than one time. We encode non-stuttering into Boolean constraints as follows:

$$\forall i, j, k : b_{ikj} = 1 \implies b_{jkj} = 1. \tag{9}$$

The utility of this constraint is demonstrated empirically in Section 5.1.

## 3.2 Algorithm

To learn a minimal reward machine from the depth-$l$ restriction of a prefix tree policy, we proceed as follows.

We start with one node and increase the number $n$ of nodes until the following SAT problem is feasible when instantiated for a graph with $n$ nodes:

$$\text{SAT}_n((5), (6), (9), \text{ for all } \{\sigma, \sigma'\} \in \mathcal{E}^-(8)). \tag{SAT}$$

By construction, this is guaranteed to be satisfiable for some $n \leq u_{\max}$, upon which a reward machine model $\mathcal{G}^{\text{learned}}$ can be constructed from the satisfying assignment's $B_k$'s. Then, we compute the product MDP model $(\mathcal{M}/r)^{\text{learned}} = \mathcal{G}^{\text{learned}} \times \mathcal{M}_L$. The learned product policy is constructed as follows. For each length $l$ word $\sigma \in (\text{AP})^l$ and for all $s \in \text{Reach}(\sigma)$, we define

$$\pi_{\text{Prod}}^{\text{learned}}(a|s, \delta_{\mathbf{u}}^{\text{learned},*}(u_I, \sigma)) = \pi_{\text{PTP}}^{\text{true},l}(a|s, \sigma), \tag{10}$$

where the transition function $\delta_{\mathbf{u}}^{\text{learned},*}$ is the transition function of $\mathcal{G}^{\text{learned}}$.

The last step is finding the numerical values of rewards that render the product policy $\pi_{\text{Prod}}^{\text{learned}}$ optimal for the product MDP model $(\mathcal{M}/r)^{\text{learned}}$. This is a standard IRL problem without reward machines where the special structure of the rewards on the product can be represented as features. We solve this step using the method developed in (Shehab et al., 2024). Featurization (see, (Shehab et al., 2024, Section 4)) is used to enforce that the reward function of the product can be written as $r((s, u), a, (s', u')) = \delta_{\mathbf{r}}(u, L(s'))$, with $u' = \delta_{\mathbf{u}}(u, L(s'))$. This IRL method gives us the corresponding output function $\delta_{\mathbf{r}}$ of the reward machine.

**Remark 1.** *Although we constrain the output function of the reward machine to be of the form $\delta_r : \mathcal{U} \times \text{AP} \rightarrow \mathbb{R}$ (leading to what is commonly known as simple reward machines (Icarte et al., 2018)), this does not limit the generality of our framework. The same procedure remains applicable in the case of a dense output function of the form $\delta_r : \mathcal{U} \times \text{AP} \rightarrow [\mathcal{S} \times \mathcal{A} \times \mathcal{S} \rightarrow \mathbb{R}]$. We focus on the simple output function formulation in this paper to simplify both the presentation and the interpretation of the resulting reward machines.*

The overall procedure is summarized in Algorithm 1.

---

**Algorithm 1:** Learning a Minimal Reward Machine from depth-$l$ Restriction of a Prefix Tree Policy

---

**Input:** Depth-$l$ prefix tree policy $\pi_{\text{PTP}}^{\text{true},l}$, labeled MDP $\mathcal{M}_L$.
**Output:** Learned reward machine $\mathcal{R}^{\text{learned}}$

**1** $n \leftarrow 1$
**2** **while** *$SAT_n$ is infeasible* **do**
**3** $\quad \lfloor \ n \leftarrow n + 1$
**4** Construct $\mathcal{E}_l^-$ using (7)
**5** $\{B_k\}_{k=1}^{|\text{AP}|} \leftarrow \text{SAT}_n$ solution
**6** $\mathcal{G}^{\text{learned}} \leftarrow \text{Construct\_RM\_model}(\{B_k\}_{k=1}^{|\text{AP}|})$
**7** $(\mathcal{M}/r)^{\text{learned}} \leftarrow \mathcal{G}^{\text{learned}} \times \mathcal{M}_L$
**8** **foreach** $\sigma \in (\text{AP})^l$ **do**
**9** $\quad$ **foreach** $s \in \text{Reach}(\sigma)$ **do**
**10** $\quad \quad \lfloor$ Define product policy: $\pi_{\text{Prod}}^{\text{learned}}(a|s, \delta_{\mathbf{u}}^{\text{learned},*}(u_I, \sigma)) \leftarrow \pi_{\text{PTP}}^{\text{true},l}(a|s, \sigma)$
**11** $\delta_{\mathbf{r}} \leftarrow \text{IRL\_to\_extract\_reward}(\pi_{\text{Prod}}^{\text{learned}}, (\mathcal{M}/r)^{\text{learned}})$
**12** **return** $\mathcal{R}^{\text{learned}} = (\mathcal{G}^{\text{learned}}, \delta_{\mathbf{r}})$

---

**Remark 2.** *It is important to emphasize that up to line 7 in Algorithm 1, neither the optimality of the prefix tree policy nor the transition kernel $\mathcal{P}$ of the MDP is used. That is, our method learns a reward machine model $\mathcal{G}^{\text{learned}}$ in a model-free fashion as long as the prefix-tree policy is induced by a reward machine. This is further illustrated through an example in Appendix C. The optimality with respect to the MaxEnt objective in (1) and the transition kernel only comes into play to extract the numerical reward values in lines 8-11.*

### 3.3 Proof of Correctness

Let $\mathcal{G}^{\text{learned}}$ be the reward machine model extracted from the SAT solution, with $\delta_{\mathbf{u}}^{\text{learned}}$ being the associated transition function. The first property of our SAT solution is that it is consistent with any fixed depth-of the prefix tree policy. We formalize this in the result below.

**Proposition 1.** *Given the labeled MDP model $\mathcal{M}_L$, the depth-$l$ restriction $\pi_{\text{PTP}}^{\text{true},l}$ of the true prefix policy $\pi_{\text{PTP}}^{\text{true}}$, and an upper bound $u_{\max}$ on the number of nodes of the underlying reward machine, let $\mathcal{G}^{\text{learned}}$ be the output of our SAT problem, and define $\pi_{\text{PTP}}^{\text{learned}}$ to be the (infinite depth) prefix tree policy induced by $\mathcal{G}^{\text{learned}}$. Then, the learned and the true prefix tree policies have the same depth-$l$ restriction, i.e., $\pi_{\text{PTP}}^{\text{learned},l} = \pi_{\text{PTP}}^{\text{true},l}$.*

*Proof.* See Appendix A.1. $\qquad\qquad\square$

While Proposition 1 represents a desirable property of our algorithm, being consistent with the depth-$l$ restriction of the true prefix tree policy is in general not sufficient to be consistent with the true (infinite-depth) prefix tree policy (as required in Problem (P1) from Section 2.4). This is potentially problematic if the agent demonstrates unseen changes in its policy for prefixes longer than $l$. At the same time, it is not possible to run our algorithm with the unrestricted prefix tree policy $\pi_{\text{PTP}}^{\text{true}}$ because it would lead to an infinite number of negative examples, i.e., $|\mathcal{E}_{\infty}^{-}| = \infty$. Consequently, the algorithm would not terminate. Fortunately, we can show that if $l$ is large enough, then increasing $l$ will not change the satisfying assignments of the SAT problem.

**Proposition 2.** *Given $\mathcal{M}_L$, an upper bound $u_{\max}$ on the number of nodes of the underlying reward machine, and the depth-$l$ restriction $\pi_{\text{PTP}}^{l}$ of some prefix tree policy $\pi_{\text{PTP}}$, where $l = |\mathcal{S}|u_{\max}^2$. Then, $\{B_k\}_{k=1}^{|\text{AP}|}$ is a satisfying assignment for*

$$\text{SAT}_{u_{\max}}((5), (6), (9), \text{ for all } \{\sigma, \sigma'\} \in \mathcal{E}_l^{-}(8))$$

*if and only if it is a satisfying assignment for all $j \geq l$ for*

$$\text{SAT}_{u_{\max}}((5), (6), (9), \text{ for all } \{\sigma, \sigma'\} \in \mathcal{E}_j^{-}(8)).$$

*Proof.* See Appendix A.2. $\qquad\qquad\square$

Now, we present the main result of this section. Our result guarantees that given a sufficiently deep restriction of the true prefix tree policy, our recovered reward machine will be consistent with true infinite depth prefix tree policy. That is, our algorithm is guaranteed to find a reward machine that is policy-equivalent to the true reward machine.

**Theorem 1.** *Given a labeled MDP model $\mathcal{M}_L$ and the depth-$l$ restriction $\pi_{\text{PTP}}^{\text{true},l}$ of a prefix tree policy induced by a reward machine $\mathcal{R}^{\text{true}}$ with at most $u_{\max}$ nodes, if $l \geq |\mathcal{S}|u_{\max}^2$, then the reward machine $\mathcal{R}^{\text{learned}}$ returned by Algorithm 1 is policy-equivalent to $\mathcal{R}^{\text{true}}$ with respect to $\mathcal{M}_L$.*

*Proof.* Follows immediately from Propositions 1 and 2. In particular, we know that $\mathcal{G}^{\text{learned}}$ is a solution of $\text{SAT}_{u_{\max},j}$, for all $j \geq l$, due to Proposition 2. Combined with Proposition 1, this means that $\pi_{\text{PTP}}^{\text{learned},j} = \pi_{\text{PTP}}^{\text{true},j}$, for all $j \geq l$. $\qquad\qquad\square$

**Remark 3.** *Note that, in practice, a depth-$l$ restriction where $l \ll |\mathcal{S}|u_{\max}^2$ can be sufficient to find a reward machine that is policy equivalent to the true one if all the solutions of the corresponding SAT problem are policy-equivalent to each other (e.g., they correspond to the same reward machine up to renaming of nodes). This will be further illustrated in the experiments of Section 5.1.*

## 4 Learning From Demonstrations

In this section, we present how our method can be applied when the optimal policy is known only through a set of optimal demonstrations $\mathfrak{D} = \{(\sigma_i, s_i, a_i)\}_{i=1}^{|\mathfrak{D}|}$. Hence, the depth-$l$ restriction of the true prefix tree

policy is unknown. Consequently, we construct the following consistent unbiased estimate:

$$\hat{\pi}_{\text{PTP}}(a|s,\sigma) = \frac{\sum_{i=1}^{|\mathfrak{D}|} \mathbb{I}(\sigma_i = \sigma, s_i = s, a_i = a)}{\sum_{i=1}^{|\mathfrak{D}|} \mathbb{I}(\sigma_i = \sigma, s_i = s)}. \tag{11}$$

However, the fact that we only have access to an approximation of $\pi_{\text{PTP}}$ leads to two challenges. First, using $\hat{\pi}_{\text{PTP}}$ to directly construct the set of negative examples $\mathcal{E}_l^-$ will lead to many pairs $\{\sigma, \sigma'\}$ being incorrectly considered as negative examples (see equation (7)). Second, the SAT problem is not robust to incorrect negative examples. In the next subsections, we address these two problems.

## 4.1 Estimating Negative Examples

To limit the inclusion of incorrect negative examples in $\mathcal{E}_l^-$, a pair of prefixes $\{\sigma_1, \sigma_2\}$ will be considered as a negative example only if we have high confidence that they correspond to negative examples given the true prefix tree policy. In particular, let $\sigma_1, \sigma_2$ be two prefixes from $\mathfrak{D}$ that we want to compare and let $s \in \mathcal{S}$ be a state. For $j \in \{1, 2\}$, let $n_j$ be the number of visitations to the pair $(s, \sigma_j)$, i.e., the denominator in (11). Note that $\pi_{\text{PTP}}(a|s, \sigma_j)$ is a categorical distribution of which we have a sample estimate for. Hence, it follows from (Weissman et al., 2003) that for all $\epsilon > 0$:

$$\mathbb{P}(\|\hat{\pi}_{\text{PTP}}(a|s,\sigma_j) - \pi_{\text{PTP}}(a|s,\sigma_j)\|_1 \le \epsilon) \ge 1 - \delta_j, \tag{12}$$

with $\delta_j \triangleq (2^{|\mathcal{A}|} - 2)e^{-\frac{1}{2}n_j\epsilon^2}$. We pick $\epsilon = \frac{1}{2}\|\hat{\pi}_{\text{PTP}}(a|s,\sigma_1) - \hat{\pi}_{\text{PTP}}(a|s,\sigma_2)\|_1$. Using a union bound, the probability of the two confidence intervals described in (12) (for $j \in \{1, 2\}$) to overlap is lower bounded by $1 - \delta_1 - \delta_2$. Consequently, we consider the pair $\{\sigma_1, \sigma_2\}$ as being a negative example if there is a state $s$ such that $1 - \delta_1 - \delta_2 \ge 1 - \alpha$, where $\alpha$ is a user-defined parameter.

## 4.2 MAX-SAT: SAT with Robustness to Incorrect Negative Examples

Even when the set of negative examples is constructed from pairs of prefixes for which we have a high confidence of being true negative examples, some pairs may still be mislabeled as negative examples. We refer to those as false positives. To deal with these false positives, we implement a weighted MAX-SAT (Manquinho et al., 2009; Biere et al., 2009) variant of the SAT problem. Concretely, the weighted MAX-SAT problem consists of finding a Boolean assignment of the variables $b_{ikj}$ such that (i) constraints (5), (6) and (9) hold, i.e., the recovered RM is deterministic, non-stuttering, and fully specified; and (ii) the constraint (8) holds for a maximum number of pairs $\{\sigma, \sigma'\}$ in $\mathcal{E}_l^-$. In other words, (5), (6) and (9) are hard constraints whereas the constraints coming from (8) are soft constraints with equal unitary weights. This approach allows for robustness to misclassifications of negative examples. We empirically demonstrate in Section 5.3 that this weighted MAX-SAT formulation consistently infers the smallest set of consistent reward machine models, even in the presence of false positives.

# 5 Experiments

To demonstrate the generality and efficiency of our approach, we apply it to a diverse set of domains, from classical grid-based MDPs to a continuous robotic control task and a real-world biological navigation dataset. In every experiment, we fix the discount factor to $\gamma = 0.99$ and the regularization weight to $\lambda = 1.0$ when solving Problem (2), both for generating demonstration traces and for reward recovery. Our code is implemented in Python, and the Z3 library (De Moura and Bjørner, 2008) is used for solving the SAT and weighted MAX-SAT problems. To enumerate all the satisfying assignments of the SAT problem, we add a constraint that the next solution should be different every time our SAT solver finds a solution. Our implementation code is made publicly available here: https://github.com/mlshehab/learning_reward_machines.git.

## 5.1 Tabular GridWorld MDPs

Our first experiment (`patrol`) is on the running-example patrol task of Figure 1. By setting $u_{\max} = 4$, our bound from Theorem 1 is 256. However, with only a depth-6 prefix tree policy, we end up with a total of

**6** solutions. These are all the possible renamings of the true reward machine (see Figure 1b), meaning that the true reward machine is learned up-to-renaming with a depth-6 prefix tree policy. Table 1 summarizes the findings, where we also show how the non-stuttering condition of Section 3.1 helps reduce the size of the negative example set, yet still recovering the true reward machine model. While some reduction in the SAT solver time is achieved, the drastic gain is in the time required to encode all the negative examples into the SAT solver, making the overall procedure orders of magnitude faster. For the remaining experiments, non-stuttering is assumed.

| Task | Depth | $|\mathcal{S}|$ | $|\mathcal{A}|$ | $|\mathcal{E}^-|$ | SAT time (sec) |
|---|---|---|---|---|---|
| `patrol` with (9) | 6 | 16 | 4 | 3076 | 0.51 |
| `patrol` without (9) | 6 | 16 | 4 | 30573 | 1.73 |
| `patrol-hallway` | 9 | 9 | 4 | 241435 | 2.859 |

Table 1: Solution statistics for the tabular GridWorld MDP.

For our second experiment (`patrol-hallway`), we add a hallway between the rooms, as shown in Figure 1c. This is a $3 \times 3$ grid world, where the corresponding label of each room is shown. The reward machine is kept the same, and the atomic proposition H is added as a self-loop to all the nodes. With this added hallway, longer atomic proposition prefixes are required to reach all nodes of the reward machine, showing how the underlying MDP affects the required depth for learning a reward machine. For example, the shortest atomic proposition that can reach $u_3$ is now $\sigma =$ AHBHC instead of ABC as in the previous example. With a depth-9 prefix tree policy, the reward machine is again learned up-to-renaming. Additional experiments studying transferability and the sensitivity to the upper bound $u_{\max}$ can be found in appendices D and E.

### 5.2 Tabular BlockWorld MDPs

The setup for this experiment is a modified block world problem (Khodeir et al., 2023; Wolfe and Barto, 2006). There are three blocks colored green, yellow and red, as well as 3 piles. Each stacking configuration of the blocks is a state of the MDP, and the action space consists of selecting a pile to pick from and a pile to place onto. We can only grab the top block of any stack. Action outcomes are assumed deterministic. The goal in the first task (`stack`) is to stack the blocks in the ordered stacking configurations $\mathbf{st_1}, \mathbf{st_2}, \mathbf{st_3}$, shown in Figure 2a. All other states have the label $\mathbf{i}$, denoting intermediate states. The corresponding reward machine is shown in Figure 2b. If the robot stacks the blocks in the order $\mathbf{st_1} \to \mathbf{st_2} \to \mathbf{st_3}$, it gets a reward of 1.0. With a depth-10 prefix tree policy and $u_{\max} = 3$, our algorithm recovers **2** consistent reward machines, which are the true reward machine up-to-renaming. The findings are summarized in Table 2. Results obtained from a finite set of demonstrations are reported in Appendix B.1.1.

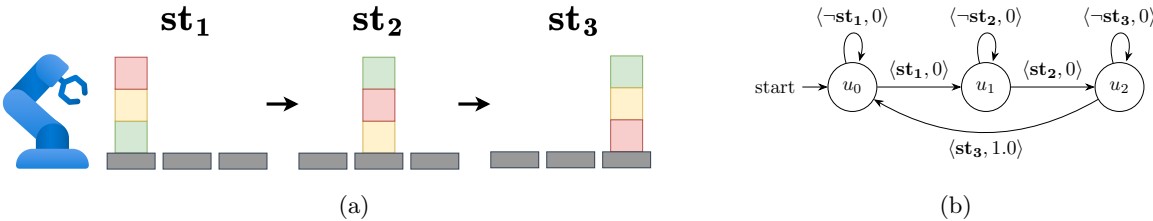

(a)                                     (b)

Figure 2: (a): Block World MDP. The left-most stacking configuration has label $\mathbf{st_1}$, where all blocks are stacked on the first pile with green being under yellow and yellow being under red. Similarly, the middle configuration has label $\mathbf{st_2}$ and the right-most configuration has label $\mathbf{st_3}$. (b): Stacking Reward Machine.

For our second task (`stack-avoid`), we introduce a "bad" state, shown in Figure 3a. The true reward machine is shown in Figure 3b. The robot's task is to stack the blocks in the order $\mathbf{st_1} \to \mathbf{st_2}$ without going through $\mathbf{st_{bd}}$. If it does so, it reaches $u_2$ and gets a reward of 1 forever. If during execution it passes through $\mathbf{st_{bd}}$, it will get a smaller (yet more immediate) reward of 0.2, but it will get stuck at $u_3$ with 0 reward forever. We note that the product policy is uniformly random in both $u_2$ and $u_3$. This means that proposition traces such as $\mathbf{st_1}, \mathbf{i}, \mathbf{st_2}$ and $\mathbf{st_1}, \mathbf{i}, \mathbf{st_{bd}}, \mathbf{st_2}$ look identical from a policy perspective, as both

reach nodes with uniformly random policies, while the first being more desirable than the second. By setting $u_{\max} = 3$, a depth-8 policy was sufficient to recover the reward machine shown in Figure 3c up-to-renaming; that is, we find a smaller reward machine consistent for this task. Notably, setting $u_{\max} = 4$ with the same depth policy yields more than a 1000 solutions, indicating that uniquely recovering the same sized reward machine -as the ground truth machine- requires longer depth policies. Results obtained from a finite set of demonstrations are reported in Appendix B.1.2.

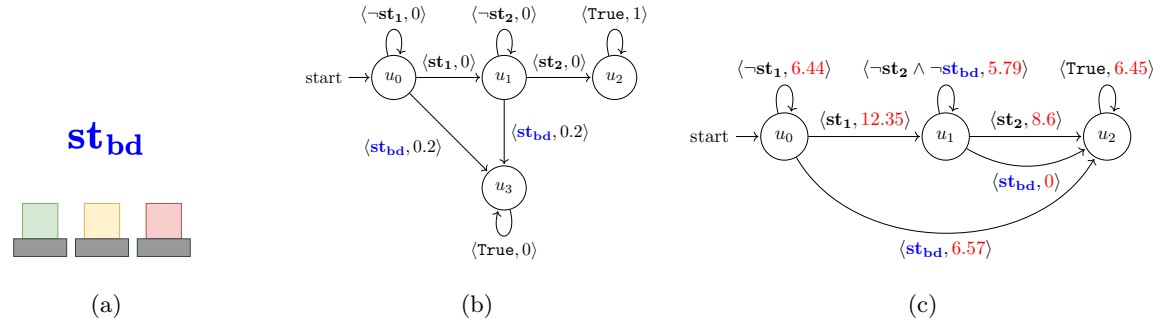

(a)          (b)          (c)

Figure 3: (a) The block stacking configuration with label $\textbf{st}_{\textbf{bd}}$ that we want our robot to avoid. (b) The ground truth reward machine. (c) Smaller consistent reward machine with the task `stack-avoid`.

| Task | Depth | $|\mathcal{S}|$ | $|\mathcal{A}|$ | $|\mathcal{E}^-|$ | SAT time (sec) |
|---|---|---|---|---|---|
| `stack` | 10 | 60 | 9 | 73548 | 0.612 |
| `stack-avoid` | 8 | 60 | 9 | 24763 | 0.569 |

Table 2: Solution statistics for the tabular BlockWorld MDP.

### 5.3 2-Link Robotic Arm with Continuous State Space

Our setup for this experiment is a modified `Reacher-v5` environment (Gymnasium (Towers et al., 2024); MuJoCo (Todorov et al., 2012)), where a planar robotic arm must reach targets randomly placed within the arena. The agent's state is a 10-D continuous vector consisting of the end-effector's position and velocity, the target's coordinates, and their mutual distance. Actions are continuous torques in $[-1, 1]^2$ applied at the elbow and shoulder joints.

Inspired by (Araki et al., 2021), we fix three colored targets — blue ($\textbf{b}$), red ($\textbf{r}$), and yellow ($\textbf{y}$) — in distinct quadrants of the arena (see Figure 4). The goal is to reach them in the order $\textbf{b} \to \textbf{y} \to \textbf{r}$. To train an expert policy, we discretize each torque dimension into five levels $\{-1.0, -0.5, 0.0, 0.5, 1.0\}$ (yielding 25 total actions). We then employ Proximal Policy Optimization (PPO) (Schulman et al., 2017), as implemented in Stable-Baselines3 (Raffin et al., 2021), to maximize a reward given by the negative Euclidean distance between the end-effector and the active target.

When generating trajectories, we switch the desired target immediately upon reaching the previous target, thereby emulating a reward machine without introducing the machine during training or simulation. Appendix B.2 provides full discretization and training details. From this procedure we collect $1M$ trajectories.

Table 3 reports, for various probability thresholds $\alpha$ (defined in Section 4.2), the size of the negative-example set $\mathcal{E}^-$ and its False Positive Rate (FPR)[1]. As expected from Section 4, solving our SAT problem with all the negative examples yields $\textbf{0}$ solutions. Instead, we employ our weighted MAX-SAT variant and recover two viable solutions (Figures 4b and 4c) up-to-renaming, which represent the best achievable given our dataset. These are the same solutions that we recover if instead we supervised with the ground-truth reward machine to remove the false positives, emphasizing the robustness of this approach.

---

[1]FPR is simply the number of false positives divided by $|\mathcal{E}^-|$.

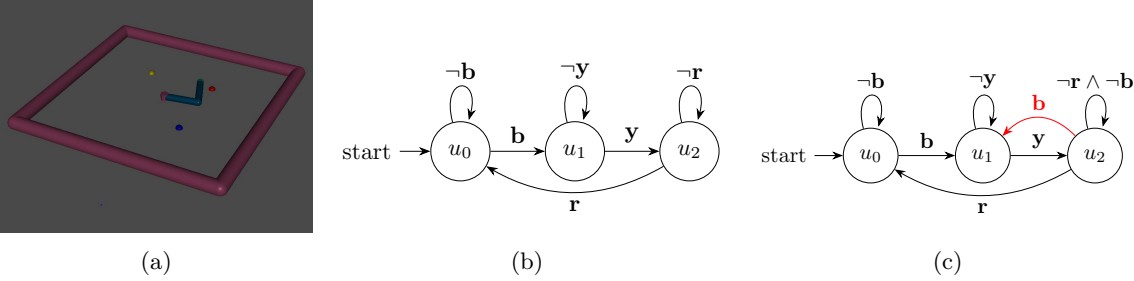

Figure 4: Reacher experiment. (a): 2-link robotic arm with the three colored targets. (b) First recovered reward machine model. (c) Second recovered reward machine model.

| $|\mathfrak{D}|$ | $|\mathcal{S}|$ | $|\mathcal{A}|$ | $|\tau|$ | $|\mathcal{E}^-|$ | $\alpha$ | FPR | # Weighted MAX-SAT solutions |
|---|---|---|---|---|---|---|---|
| 1M | 17.4M | 25 | 160 | 1472 | 0.001 | 1.90% | 4 |
| 1M | 17.4M | 25 | 160 | 1193 | 0.0001 | 1.67% | 4 |
| 1M | 17.4M | 25 | 160 | 882 | 0.00001 | 1.36% | 4 |

Table 3: Summary of the experiments for the 2D-continuous reacher arm.

## 5.4 Real-world Mice Navigation

We also applied our learning framework to the trajectories of real mice navigating (in the dark) in a 127-node labyrinth maze (Rosenberg et al., 2021) shown in Figure 5. Each node, labeled with a number in Figure 5a, represents a state in the MDP. The mouse can select from 4 actions: {stay, right, left, reverse}. State 116 (middle-right) contains a water resource and is labeled $\mathbf{w}$, state 0 (center) is the home state labeled $\mathbf{h}$, and all other states are labeled $\mathbf{i}$. Two cohorts of 10 mice moved freely in the maze for 7 hours, with one cohort being water-restricted and the other was not. A sample water-restricted mouse trajectory is shown in Figure 5b. Difference in water restriction condition resulted in different animal behavior between these two cohorts. For the purposes of our study, we only considered the water-restricted mice. We used the same dataset of trajectories from (Ashwood et al., 2022), which is comprised of 200 mouse trajectories, given as state-action pairs of length 22 each. We set our probability threshold at $\alpha = 0.001$. With $u_{\max} = 2$, our algorithm learns the unique reward machine shown in Figure 5c. This is consistent with the seen behaviors of water-restricted mice which first try to reach the water port and hydrate, after which their behaviors switch to exploring the maze or going back to the home state (Zhu et al., 2024; Ashwood et al., 2022). More details can be found in Appendix B.3.

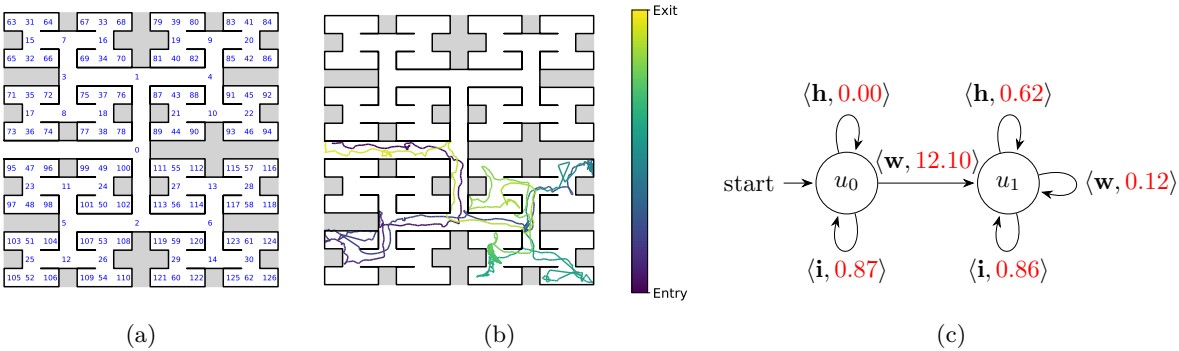

Figure 5: Labyrinth experiment. (a): Maze structure and state space definition. (b): Trajectory of a single mouse. (c): Recovered reward machine. (a) and (b) are reprinted from (Rosenberg et al., 2021).

We further evaluate the quality of the recovered reward machine using a held-out set of unseen trajectories (20 test trajectories). The reward machine model and the product policy are learned from the remaining 180 training trajectories. For each trajectory in the test set, we compute its log-likelihood under the product policy. This metric reflects how well the learned model captures the underlying trajectory distribution

and generalizes beyond the training data. The results are shown in Table 4 and represent the average log-likelihood over the test dataset. To contextualize the results, we compare against three baselines: a uniformly random policy and two variants of Max Causal Entropy IRL Ziebart (2010). **D-IRL** employs a dense feature representation, assigning a distinct reward value to every MDP state. **F-IRL** uses a structured feature representation aligned with the environment's labeling. Specifically, its feature vector is a one-hot encoding indicating whether the agent is at the home port (**h**), at the water port (**w**), or in any other intermediate state (**i**). These results also quantitatively demonstrate the superiority of the learned reward machine in capturing the unseen behaviors of the mice.

| Method | D-IRL | F-IRL | Uniform | LRM |
|---|---|---|---|---|
| **Average log-likelihood** | $-28.99$ | $-28.57$ | $-30.49$ | $\mathbf{-9.81}$ |

Table 4: Average log-likelihood performance of the learned reward machine on unseen trajectories, compared against three baseline methods.

## 6 Limitations and future work

The present framework assumes that the atomic propositions provided by the labeling function are exact, meaning no noise or mislabeling is present in the observed labels. While this assumption simplifies the analysis and ensures that the negative examples used in the SAT formulation are correct, it may be unrealistic in settings where the labeling function is derived from perception modules or noisy sensors. Future work could address this limitation by extending the framework to handle noise in the labeling function, for instance by integrating robust RM-learning techniques such as those in Parac et al. (2024), thereby broadening applicability to real-world domains.

Another limitation arises from the requirement to identify all negative examples up to depth $l = |\mathcal{S}|u_{\max}^2$ for the theoretical results to apply (see Theorem 1). Although this bound is polynomial in the size of the MDP and the maximum number of RM nodes, the number of distinct negative examples —and thus the number of clauses in the SAT formulation— may grow exponentially with $l$ in the worst case. For large problems, this can lead to SAT instances of prohibitive size and render the approach computationally intractable. On the other hand, in practice, we were able to identify the reward machine model with a much smaller depth in all our examples. Therefore, the existence of a tighter sufficient depth bound is an open problem, which we will investigate further in the future. We are also interested in developing smart prefix-selection algorithms that incorporate a verification loop to check, on the fly, whether the selected negative examples are sufficient.

## 7 Conclusion

In this work, we present a framework for learning reward machines from partially observed policies, where neither the rewards nor the reward machine states are available to the learner. Instead, our method constructs a SAT problem instance from a sufficiently deep prefix tree policy, from which a reward machine equivalent to the true underlying one can be recovered. The approach is generalized to learn the reward machines directly from demonstrations from an optimal expert, where robustness to inaccuracies in policy estimates is handled using finite-sample confidence bounds and MAX-SAT is used to eliminate incorrect negative examples. A diverse set of experiments illustrate the effectiveness of our approach.

**Acknowledgments:** This work is supported in part by ONR CLEVR-AI MURI (#N00014- 21-1-2431).

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

## Appendices

## A   Proofs

### A.1   Proof of Proposition 1

*Proof.* We proceed by contradiction. Assume that there exists some $\sigma' \in \text{Pref}(\mathcal{L}(\mathcal{M}_L))$, with $|\sigma'| \leq l$, a state $s \in \text{Reach}(\sigma')$ and an action $a \in \mathcal{A}$ such that:

$$\pi_{\text{PTP}}^{\text{learned}}(a|s, \sigma') \neq \pi_{\text{PTP}}^{\text{true}}(a|s, \sigma'). \tag{13}$$

Let $u \triangleq \delta_{\mathbf{u}}^{*,\text{learned}}(u_I, \sigma')$. The left-hand-side of (13) can be written as:

$$\pi_{\text{PTP}}^{\text{learned}}(a|s, \sigma') = \pi_{\text{Prod}}^{\text{learned}}(a|s, u) = \pi_{\text{PTP}}^{\text{true}}(a|s, \sigma), \text{ for some } \sigma \in \text{Path}(u) \cap \text{Pref}(\mathcal{L}(\mathcal{M}_L)), \text{ with } |\sigma| \leq l,$$

where the second equality is due to (10). We get:

$$\pi_{\text{PTP}}^{\text{true}}(a|s, \sigma) \neq \pi_{\text{PTP}}^{\text{true}}(a|s, \sigma'), \text{ where both } \sigma, \sigma' \in \text{Path}(u).$$

More precisely, $\delta_{\mathbf{u}}^{*,\text{learned}}(u_I, \sigma) = \delta_{\mathbf{u}}^{*,\text{learned}}(u_I, \sigma')$. Due to the contrapositive of Lemma 1, we have a contradiction. Similarly, due to the full specification condition of Section 3.1 and (10), the support of the two prefix policies will be the same by construction.   □

### A.2   Proof of Proposition 2

We start by formally defining some important concepts that will be central for proving our result. Our proof idea requires reasoning about joint (i.e. synchronized) paths over two distinct reward machine models, and being able to shrink these joint paths by removing cycles (i.e. loops). To start, we define *cycles* in a product MDP model $\mathcal{G} \times \mathcal{M}_L$ as follows:

**Definition 2.** *Given a product MDP model $\mathcal{G} \times \mathcal{M}_L$ and a proposition sequence $\sigma = l_1 \cdots l_k$, generated from a state sequence $\tau = (s_1, s_2, \cdots, s_k)$ (i.e., $\sigma \in \text{Pref}(\mathcal{L}(\mathcal{M}_L))$), we say that a subsequence $\sigma_{i:j}$ of $\sigma$ is a **cycle** in $\mathcal{G} \times \mathcal{M}_L$ if $s_i = s_j$ and $\delta_{\mathbf{u}}^*(u_I, \sigma_{:i}) = \delta_{\mathbf{u}}^*(u_I, \sigma_{:j})$.*

We will use the above definition to construct shorter label sequences with no cycles given a long label sequence. In particular, let $l_c \triangleq |\mathcal{S}||\mathcal{U}|$ be the number of states in $\mathcal{G} \times \mathcal{M}_L$. By the pigeonhole principle, we know that any state trajectory of length more than $l_c$ has to visit some product state more than once, meaning that it has at least one cycle. In particular, given any proposition sequence $\sigma$, with $|\sigma| > l_c$, let $\bar{\sigma}$ be the subsequence of $\sigma$ obtained by removing all the cycles in $\sigma$. Then, we know that $|\bar{\sigma}| \leq l_c$, since $\bar{\sigma}$ has at most all the unique states from $\sigma$, which cannot exceed $l_c$. Note that removing cycles preserves the last product state reached from following $\sigma$.

Next, we define *synchronized reward machine models*.

**Definition 3.** *Let $\mathcal{G}_1 = (\mathcal{U}_1, u_I^1, \text{AP}, \delta_{\mathbf{u}}^1)$, $\mathcal{G}_2 = (\mathcal{U}_2, u_I^2, \text{AP}, \delta_{\mathbf{u}}^2)$ be two reward machine models, with $|\mathcal{U}_1|, |\mathcal{U}_2| \leq u_{\max}$. The **synchronized reward machine model** is a reward machine model defined as follows:*

$$\mathcal{G}^{\text{sync}} = (\mathcal{U}^{\text{sync}}, u_I^{\text{sync}}, \text{AP}, \delta_{\mathbf{u}}^{\text{sync}})$$
$$\mathcal{U}^{\text{sync}} = \mathcal{U}_1 \times \mathcal{U}_2,$$
$$u_I^{\text{sync}} = (u_I^1, u_I^2),$$
$$\delta_{\mathbf{u}}^{\text{sync}}((u_i^1, u_j^2), l) = (\delta_{\mathbf{u}}^1(u_i^1, l), \delta_{\mathbf{u}}^2(u_j^2, l)), \quad l \in \text{AP}.$$

Similarly to a regular reward machine model, the product $\mathcal{G}^{\text{sync}} \times \mathcal{M}_L$ is well defined. The total number of states in $\mathcal{G}^{\text{sync}} \times \mathcal{M}_L$ is upper bounded by $l = |\mathcal{S}|u_{\max}^2$. In particular, consider an arbitrary label sequence $\sigma \in \text{Pref}(\mathcal{L}(\mathcal{M}_L))$, generated from a state sequence $\tau = (s_1, s_2, \cdots, s^*)$, with $|\sigma| > |\mathcal{M}|u_{\max}^2$,

and let $(u, u') = \delta_{\mathbf{u}}^{\text{sync},*}(u_I^{\text{sync}}, \sigma)$. This means that the synchronized product state $(u, u', s^*)$ is reachable in $\mathcal{G}^{\text{sync}} \times \mathcal{M}_L$. Thus, by removing cycles of $\sigma$ in $\mathcal{G}^{\text{sync}} \times \mathcal{M}_L$, we can construct a shorter prefix $\bar{\sigma}$, with $|\bar{\sigma}| \leq |\mathcal{M}| u_{\max}^2$, such that $(u, u') = \delta_{\mathbf{u}}^{\text{sync},*}(u_I^{\text{sync}}, \bar{\sigma})$, and $s^*$ is the MDP state reached.

We are now ready for the proof of Proposition 2. We provide Figure 6 as an illustration of the proof.

*Proof.* For a shorthand notation, denote the first SAT instance in the proposition statement as $\text{SAT}_{u_{\max}, l}$ and the second SAT instance as $\text{SAT}_{u_{\max}, j}$. Also note that $\mathcal{E}_l^- \subseteq \mathcal{E}_j^-$ when $j \geq l$ as the negative examples can only grow as the depth increases. Throughout the proof, we interchange $\pi_{\text{PTP}}^j$ and $\pi_{\text{PTP}}$ as both are equal up to depth $j$.

We need to show that the additional negative examples in $\mathcal{E}_j^- \setminus \mathcal{E}_l^-$ do not change the set of satisfying assignments. The $\impliedby$ direction is straightforward, since a satisfying assignment cannot become unsatisfying by removing constraints.

For the $\implies$ direction, take a satisfying assignment $\{B_k\}_{k=1}^{|\text{AP}|}$ for $\text{SAT}_{u_{\max}, l}$ and assume by contradiction that $\{B_k\}_{k=1}^{|\text{AP}|}$ is not satisfying for $\text{SAT}_{u_{\max}, j}$ with $j > l$. Consider the reward machine model $\mathcal{G}^{\text{learned}}$, with the transition function $\delta_{\mathbf{u}}^{\text{learned}}$, corresponding to $\{B_k\}_{k=1}^{|\text{AP}|}$. Since $\{B_k\}_{k=1}^{|\text{AP}|}$ is unsatisfying for $\text{SAT}_{u_{\max}, j}$, then there exists a negative example $\{\sigma, \sigma'\} \in \mathcal{E}_j^- \setminus \mathcal{E}_l^-$ such that

$$\delta_{\mathbf{u}}^{\text{learned},*}(u_I, \sigma) = \delta_{\mathbf{u}}^{\text{learned},*}(u_I, \sigma'), \tag{14}$$

while $\pi_{\text{PTP}}^j(a^*|s^*, \sigma) \neq \pi_{\text{PTP}}^j(a^*|s^*, \sigma')$ for some $(s^*, a^*) \in \mathcal{S} \times \mathcal{A}$. These two facts are shown as ${\color{green}\neq_{\pi_{\text{PTP}}}}$ and ${\color{green}=_u}$ connecting $\sigma$ and $\sigma'$ in Figure 6.

Now, let $\mathcal{G}^{\text{true}} = (\mathcal{U}, u_I, \text{AP}, \delta_{\mathbf{u}}^{\text{true}})$ be a reward machine model consistent with $\pi_{\text{PTP}}$. We define the following nodes, along with the associated product states:

$$u^{\text{true}} \triangleq \delta_{\mathbf{u}}^{\text{true}}(u_I, \sigma), \quad (u^{\text{true}}, s^*) \in \mathcal{G}^{\text{true}} \times \mathcal{M}_L,$$
$$u^{\text{learned}} \triangleq \delta_{\mathbf{u}}^{\text{learned}}(u_I, \sigma), (u^{\text{learned}}, s^*) \in \mathcal{G}^{\text{learned}} \times \mathcal{M}_L,$$
$$u^{\text{true},\prime} \triangleq \delta_{\mathbf{u}}^{\text{true}}(u_I, \sigma'), \quad (u^{\text{true},\prime}, s^*) \in \mathcal{G}^{\text{true}} \times \mathcal{M}_L,$$
$$u^{\text{learned},\prime} \triangleq \delta_{\mathbf{u}}^{\text{learned}}(u_I, \sigma'), (u^{\text{learned},\prime}, s^*) \in \mathcal{G}^{\text{learned}} \times \mathcal{M}_L.$$

Hence, by (14), we have that $u^{\text{learned}} = u^{\text{learned},\prime}$. Let $\mathcal{G}^{\text{sync}}$ be the synchronized reward machine model between $\mathcal{G}^{\text{learned}}$ and $\mathcal{G}^{\text{true}}$ according to Definition 3. We observe the following:

$$(u^{\text{true}}, u^{\text{learned}}) = \delta_{\mathbf{u}}^{\text{sync},*}(u_I^{\text{sync}}, \sigma), \quad (u^{\text{true}}, u^{\text{learned}}, s^*) \in \mathcal{G}^{\text{sync}} \times \mathcal{M}_L,$$
$$(u^{\text{true},\prime}, u^{\text{learned},\prime}) = \delta_{\mathbf{u}}^{\text{sync},*}(u_I^{\text{sync}}, \sigma'), \quad (u^{\text{true},\prime}, u^{\text{learned},\prime}, s^*) \in \mathcal{G}^{\text{sync}} \times \mathcal{M}_L.$$

This means that the synchronized product states $(u^{\text{true}}, u^{\text{learned}}, s^*)$ and $(u^{\text{true},\prime}, u^{\text{learned},\prime}, s^*)$ are both reachable in $\mathcal{G}^{\text{sync}} \times \mathcal{M}_L$. Thus, by removing cycles, there must exist shorter sequences, $\bar{\sigma}, \bar{\sigma}'$, with $|\bar{\sigma}|, |\bar{\sigma}'| \leq |\mathcal{S}| u_{\max}^2$, such that:

$$(u^{\text{true}}, u^{\text{learned}}) = \delta_{\mathbf{u}}^{\text{sync},*}(u_I^{\text{sync}}, \bar{\sigma}),$$
$$(u^{\text{true},\prime}, u^{\text{learned},\prime}) = \delta_{\mathbf{u}}^{\text{sync},*}(u_I^{\text{sync}}, \bar{\sigma}'). \tag{15}$$

Note that $s^*$ is still the reached MDP state in both synchronized product nodes above. By the definition of $\delta_{\mathbf{u}}^{\text{sync},*}$, we can decompose (15) into:

$$u^{\text{true}} = \delta_{\mathbf{u}}^{\text{true}}(u_I, \bar{\sigma}), \ u^{\text{learned}} = \delta_{\mathbf{u}}^{\text{learned}}(u_I, \bar{\sigma}),$$
$$u^{\text{true},\prime} = \delta_{\mathbf{u}}^{\text{true}}(u_I, \bar{\sigma}'), \ u^{\text{learned},\prime} = \delta_{\mathbf{u}}^{\text{learned}}(u_I, \bar{\sigma}'). \tag{16}$$

This means that $\sigma$ and $\bar{\sigma}$ lead to the same node $u^{\text{true}}$ in $\mathcal{G}^{\text{true}}$. Similarly, $\sigma'$ and $\bar{\sigma}'$ both lead to the same node $u^{\text{true},\prime}$. Since $\mathcal{G}^{\text{true}}$ is consistent with $\pi_{\text{PTP}}$, the following holds:

$$\pi_{\text{PTP}}(a|s^*, \bar{\sigma}) = \pi_{\text{PTP}}(a|s^*, \sigma), \quad \forall a \in \mathcal{A},$$
$$\pi_{\text{PTP}}(a|s^*, \bar{\sigma}') = \pi_{\text{PTP}}(a|s^*, \sigma'), \quad \forall a \in \mathcal{A}. \tag{17}$$

Since $\pi_{\text{PTP}}(a^*|s^*, \sigma) \neq \pi_{\text{PTP}}(a^*|s^*, \sigma')$ due to our contradiction assumption, we conclude from (17) that $\pi_{\text{PTP}}(a^*|s^*, \bar{\sigma}) \neq \pi_{\text{PTP}}(a^*|s^*, \bar{\sigma}')$. However, $u^{\text{learned}} = u^{\text{learned},\prime}$ combined with (16) implies that $\delta_{\mathbf{u}}^{\text{learned}}(u_I, \bar{\sigma}) = \delta_{\mathbf{u}}^{\text{learned}}(u_I, \bar{\sigma}')$, contradicting that $\{B_k\}_{k=1}^{|\text{AP}|}$ is a SAT assignment for the depth $l$. This concludes the proof. $\square$

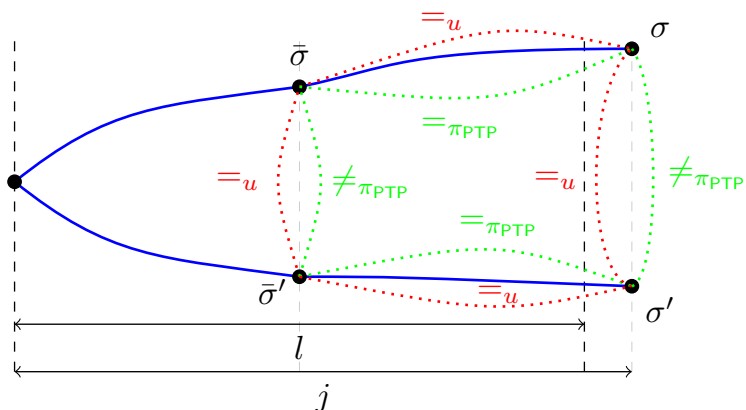

Figure 6: Proof Illustration of Proposition 2. $=_{\pi_{\text{PTP}}}, \neq_{\pi_{\text{PTP}}}$ means that the prefix tree policy $\pi_{\text{PTP}}$ is equal/different for the corresponding sequences. $=_u$ means that the corresponding sequences arrive at the same node in $\mathcal{G}^{\text{learned}}$.

# B    Experiment Details and Additional Results

## B.1    Tabular BlockWorld MDP

To generate the expert trajectory dataset, we first compute the product policy given the BlockWorld MDP dynamics and the corresponding ground-truth reward machine. We use soft bellman iteration to find the optimal product policy. The procedure is summarized in Algorithm 2.

At each time step, we use the visited proposition prefix $\sigma$ to find the reached node $u$ on the true reward machine, using $u = \delta_{\mathbf{u}}^*(u_I, \sigma)$. We then sample an action from the true product policy $\pi^*(a|s, u)$, where $s$ is the MDP state reached. We keep count of the sampled actions at all state-prefix pairs for constructing the prefix tree policy. The procedure is summarized in Algorithm 3. In the following section, we show the results for the `stack` and `stack-avoid` tasks.

### B.1.1    (`stack`) task

We simulated varying size demonstration datasets and solved our SAT problem with each. In Table 5, we show the effect of the number of demonstrations in reducing the number of satisfying solutions. With a demonstration dataset of size $1M$, the ground-truth (`stack`) reward machine is recovered up-to-renaming. However, even a demonstration set of size 5000 can indeed reduce the number of satisfying solutions to 4. These 4 solutions include the ground-truth reward machine (accounting for 2 solutions due to renaming), and the reward machine (up-to-renaming again) shown in Figure 7b, where $\delta_{\mathbf{u}}(u_2, \mathbf{a}) = u_1$, instead of $\delta_{\mathbf{u}}(u_2, \mathbf{a}) = u_2$ as in the ground-truth reward machine.

### B.1.2    (`stack-avoid`) task

In Table 6, we similarly show the effect of the size of the simulated dataset on the number of satisfying solutions for the `stack-avoid` task. In this task, $1M$ trajectories were not enough to extract all the necessary negative examples to reduce the number of solutions to 2. However, a much smaller dataset of size 1000

---

**Algorithm 2:** Soft Bellman Iteration on the Product MDP

---

**Input:** MDP $\mathcal{M}$, ground-truth reward machine $\mathcal{R}$, labeling function $L$, tolerance $\epsilon$.

**Output:** Optimal product policy $\pi^*$

**1** Construct product MDP $\mathcal{M}_{\text{Prod}} = \left(\mathcal{S}', \, \mathcal{A}', \, \mathcal{P}', \, \mu_0', \, \gamma', \, r'\right)$ as in Section 2.2.

**2** Initialize

$$V_0(s, u) \leftarrow 0 \quad \forall (s, u) \in \mathcal{S}', \quad \text{error} \leftarrow \infty, \quad k \leftarrow 0$$

**3** **while** *error* $> \epsilon$ **do**

**4**    **foreach** $(s, u) \in \mathcal{S}'$ **do**

**5**       **foreach** $a \in \mathcal{A}'$ **do**

**6**          $Q_{k+1}(s, u, a) \leftarrow r'\big((s, u), a\big) + \gamma' \sum_{(s', u') \in \mathcal{S}'} \mathcal{P}'\big((s, u), a, (s', u')\big) V_k(s', u')$

**7**       $V_{k+1}(s, u) \leftarrow \log \sum_{a \in \mathcal{A}'} \exp\big(Q_{k+1}(s, u, a)\big)$

**8**    error $\leftarrow \max_{(s, u) \in \mathcal{S}'} \big|V_{k+1}(s, u) - V_k(s, u)\big|$

**9**    $k \leftarrow k + 1$

**10** **foreach** $(s, u) \in \mathcal{S}', \, a \in \mathcal{A}'$ **do**

**11**    $\pi^*(a \mid s, u) \leftarrow \exp\big(Q_k(s, u, a) - V_k(s, u)\big)$

**12** **return** $\pi^*$

---

**Algorithm 3:** Constructing the Prefix-Tree Policy via Simulation

---

**Input:** MDP $\mathcal{M}$, reward machine $\mathcal{R}$, true product policy $\pi^*(a \mid s, u)$, # trajectories $N$, trajectory length $H$

**Output:** Prefix-tree policy $\pi_{\text{PTP}}^H$

**1** Initialize counts: $C(\sigma, s, a) \leftarrow 0$ for all prefixes $\sigma$, states $s \in \mathcal{S}$, actions $a \in \mathcal{A}$.

**2** **for** $i \leftarrow 1$ **to** $N$ **do**

**3**    $\sigma \leftarrow \epsilon$   (empty prefix)

**4**    $s \sim \mu_0$   (initial MDP state)

**5**    $u \leftarrow u_I$   (initial RM node)

**6**    **for** $t \leftarrow 1$ **to** $H$ **do**

**7**       $u \leftarrow \delta_{\text{u}}(u, \sigma)$;

**8**       Sample $a \sim \pi^*(\cdot \mid s, u)$;

**9**       Execute $a$, observe $s' \sim \mathcal{P}(s, a)$ and label $\ell \leftarrow L(s')$;

**10**       $\sigma \leftarrow \sigma \parallel \ell$   (concatenation);

**11**       $C(\sigma, s, a) \leftarrow C(\sigma, s, a) + 1$;

**12**       $s \leftarrow s'$;

**13** **foreach** *prefix $\sigma$, state $s$, action $a$* **do**

**14**    $\pi_{\text{PTP}}^H(a \mid s, \sigma) = \dfrac{C(\sigma, s, a)}{\sum_{a' \in \mathcal{A}} C(\sigma, s, a')}$.

**15** **return** $\pi_{\text{PTP}}^H$

---

| $|\mathfrak{D}|$ | $|\tau|$ | $|\mathcal{E}^-|$ | $\alpha$ | # SAT solutions |
|---|---|---|---|---|
| 1000 | 20 | 58 | 0.05 | 24 |
| 3000 | 20 | 174 | 0.05 | 12 |
| 5000 | 20 | 305 | 0.05 | 8 |
| 10000 | 20 | 490 | 0.05 | 8 |
| 0.1M | 20 | 2249 | 0.05 | 4 |
| 1M | 20 | 7995 | 0.05 | 2 |

Table 5: Size of the simulated dataset vs. the number of satisfying solutions.

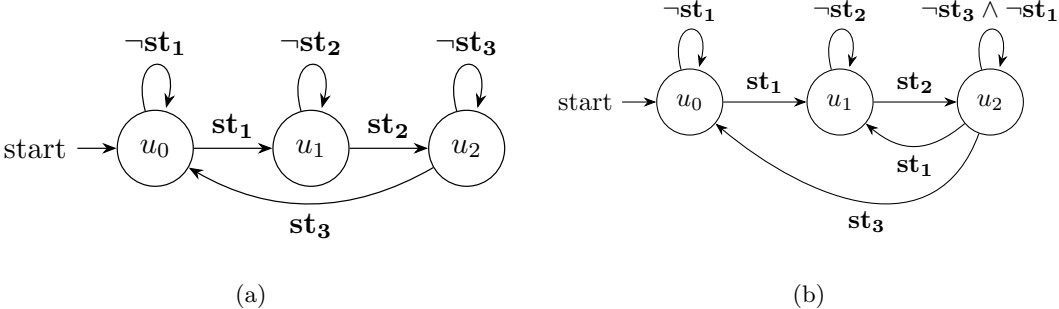

(a)                                                    (b)

Figure 7: Recovered Reward Machine Models for `stack` task.

was enough to reduce the number of solutions to **4**, which we can manually inspect. These solutions are shown in Figures 8a and 8b (up-to-renaming). The reward machine in Figure 8b differs from the ground-truth reward machine by assigning $\delta_u(u_1, \mathbf{st_{bd}}) = u_1$, instead of $\delta_u(u_1, \mathbf{st_{bd}}) = u_2$. Such an ambiguity could have been resolved by a negative example of the form $\{\sigma_1 = \mathbf{st_1}, \mathbf{i}, \mathbf{st_1}, \sigma_2 = \mathbf{st_1}, \mathbf{i}, \mathbf{st_{bd}}, \mathbf{i}, \mathbf{st_1}\}$, which the MDP is capable of producing given the length of the simulated trajectories. However, since the policy becomes uniformly random after reaching $\mathbf{st_{bd}}$, simulating such negative examples might require a much larger dataset.

| $|\mathfrak{D}|$ | $|\tau|$ | $|\mathcal{E}^-|$ | $\alpha$ | # SAT solutions |
|---|---|---|---|---|
| 200 | 20 | 26 | 0.05 | 32 |
| 500 | 20 | 52 | 0.05 | 8 |
| 1000 | 20 | 67 | 0.05 | 8 |
| 0.1M | 20 | 1615 | 0.05 | 4 |
| 1M | 20 | 5330 | 0.05 | 4 |

Table 6: Size of the simulated dataset vs. the number of satisfying solutions.

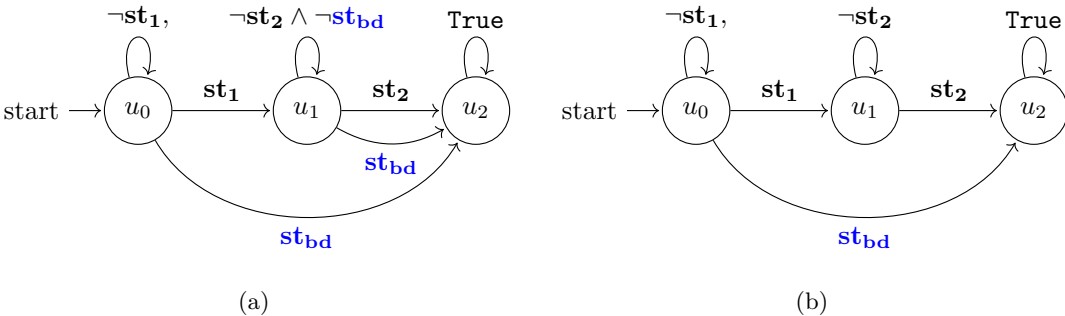

(a)                                                    (b)

Figure 8: Recovered Reward Machine Models for `stack-avoid` task.

### B.2 2-Link Reacher Robotic Arm with Continuous State Space

#### B.2.1 Expert policy training using deep RL

To train the policy, we used the standard `Reacher-v5` environment from Gymnasium, augmented with custom wrappers to discretize the action space and introduce randomized initial conditions. We also increased the episode horizon from the default 50 steps to 160 steps in order to match the expected time to visit all 3 desired poses and finish the task. The `DiscreteReacherActionWrapper` maps the continuous two-dimensional action space to a `MultiDiscrete` space of 25 torque combinations by restricting each component to one of five values: {-1.0, -0.5, 0.0, 0.5, 1.0}. Additionally, the `ForceRandomizedReacher` wrapper introduces randomized initial joint positions and velocities at each episode reset to promote robustness and generalization. This modification was essential, as the default initial conditions in the base environment are typically limited to small perturbations around fully-extended arm configurations. Training solely from such narrow initializations led to poor task performance and limited the policy's ability to generalize across the broader state space.

We trained a Proximal Policy Optimization (PPO) agent using the `stable-baselines3` library, with all the default settings and hyperparameters. To accelerate training, we employed a vectorized environment with 50 parallel instances running on CPU. The agent was trained for 20 million time steps using a multilayer perceptron (MLP) policy.

#### B.2.2 Discretizing the state space and sampling the policy

The MDP state space is generated by discretizing the shoulder angle $\theta_1$, the elbow angle $\theta_2$, and the angular velocities $\dot{\theta}_1, \dot{\theta}_2$. The angles are discretized uniformly in $[-\pi, \pi]$ with a bin size of $10° \approx 0.17$ radians. The angular velocities are discretized uniformly in $[-14, 14]$ with a bin size of 0.25 rad/s. This leads to $\approx 17.4M$ states in the MDP.

When sampling an action from the optimal policy, we first identify the discrete state corresponding to the current continuous state. The action is then sampled at the center point of that discrete region. This approach standardizes the policy's behavior across different continuous states that map to the same discrete representation, ensuring consistency within each discretized region of the state space.

### B.3 Real-world Mice Navigation

For recovering the reward function, we first need to construct the product policy. However, real-world data generally do not perfectly fit the assumed mathematical models, i.e., the max entropy assumption. Since most of the states in the MDP are unvisited, and many action are not sampled at most of the states, the product policy is predominantly sparse ($\sim 92\%$ of the action probabilities are 0). We preprocess the policy by clipping its minimum value to 0.05. This allows us to use the results from (Shehab et al., 2024) to compute the reward function, as we require computing the log of the policy. The reward space is constrained to satisfy the feature-decomposition which the reward machine provides. However, there is generally no guarantee that we can find such a featurized reward function that can perfectly induce any product policy. With the shown reward in Figure 5c, the norm difference between the induced policy and the true policy is 2.66 (the norm difference with a uniformly random policy is 2.97). By increasing $u_{\max}$ to 3, one of the recovered reward machines is shown in Figure 9, which yields a slightly larger policy difference norm of 2.71.

## C Policy Optimality is not Required

In this section, we illustrate with a toy example that the optimality of the prefix tree policy is not necessary to recover the reward machine model. We consider the MDP given by the BlockWorld of Section 5.2, and a simple 2-node reward machine model given by Figure 10. We do not specify a reward function here nor solve the forward MaxEnt RL problem before simulating the dataset, and instead specify an arbitrary stochastic policy at each node of the reward machine. With a dataset size $50K$ and trajectory length 20, our method learns uniquely the reward machine model.

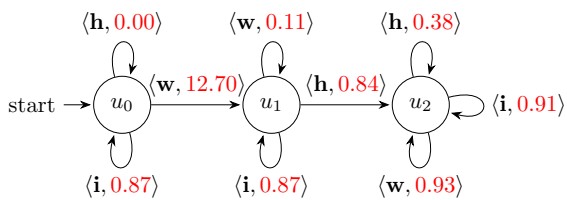

Figure 9: Recovered Reward machine for Mice Dataset with $u_{\max} = 3$.

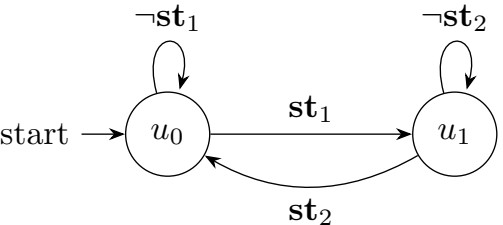

Figure 10: Reward machine model considered in Appendix C.

## D Sensitivity to the Node Bound $u_{\max}$

To understand the effect of misspecifying the number of nodes on the resulting learned reward machine, we run an additional experiment in the Gridworld benchmark (see Section 5.1) while varying the bound on the number of nodes. We simulate a dataset of $2,500$ trajectories of length 10 each. We then construct the negative example set using our finite sample results and solve for a reward machine model using MaxSAT. The recovered models are shown in Figure 11 below. These reward machine models still qualitatively capture the overall task even with the limited number of nodes. To quantitatively assess the quality of these models, we first learn a product policy for each using Algorithm 4 below. We then perform trajectory rollouts to compute the return of the policy consistent with each model. While the actions are sampled according to the corresponding learned policy of each model, the reward is calculated based on the traversed nodes in the ground-truth reward machine, given by Figure 1b[2]. Results in Table 7 show a graceful degradation in performance as the number of reward machine nodes decreases. We also implemented a Max Entropy IRL baseline Ziebart et al. (2008), where a static reward is learned from the demonstration set.

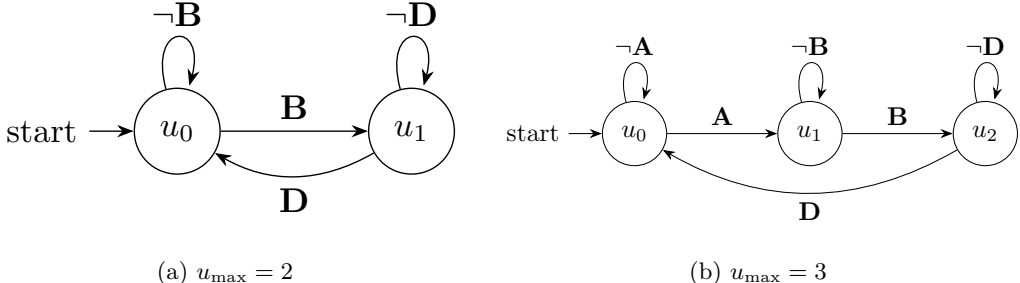

(a) $u_{\max} = 2$         (b) $u_{\max} = 3$

Figure 11: Recovered reward machine models with varying node bounds.

.

| Model | Total # clauses | # used clauses | Rollouts | $|\tau|$ | Average Return per Rollout |
|---|---|---|---|---|---|
| $u_{\max} = 4$ | 1112 | 1112 | 10K | 100 | 176.62 |
| $u_{\max} = 3$ | 1112 | 902 | 10K | 100 | 148.74 |
| $u_{\max} = 2$ | 1112 | 589 | 10K | 100 | 94.25 |
| D-IRL | - | - | 10K | 100 | 17.12 |
| F-IRL | - | - | 10K | 100 | 14.49 |

Table 7: Statistics for the varying bound experiment. (Total # clauses) is the number of negative examples from the finite dataset. (# used clauses) is the number of negative examples which MaxSAT included as hard constraints. (Rollouts) is the total number of rollouts. ($|\tau|$) is the length of the trajectory. (Average Return per Rollout) is the sum of accumulated rewards in all the rollouts divided by the number of rollouts.

---

[2]For this experiment we set the reward for completing the task at 10.

---

**Algorithm 4:** Construct Learned Product Policy

---

**Input:** Learned reward machine model $\widehat{\mathcal{G}}$, Ground-truth reward machine model $\mathcal{G}$, True prefix tree
policy $\pi_{\mathrm{PTP}}$, Labeling function $L$, MDP $\mathcal{M}$, prefix length $d_p$
**Output:** Learned product policy $\widehat{\pi}$

**1 foreach** *node* $u \in \widehat{\mathcal{G}}$ **do**
**2**    Compute Reach$(u)$ ;
   // we consider prefixes up to a fixed length of $d_p$
**3**    **foreach** $\sigma \in \mathrm{Paths}(u)$ **do**
**4**       Compute Reach$(\sigma)$;
**5**       **foreach** $s \in \mathrm{Reach}(\sigma)$ **do**
**6**          **foreach** $a \in \mathcal{A}$ **do**
**7**             $\widehat{\pi}(a \mid s, u) \mathrel{+}= \pi_{\mathrm{PTP}}(a \mid s, \sigma)$

**8 Normalize** $\widehat{\pi}$ and fill in zero rows with uniform policy;
**9 return** $\widehat{\pi}$

---

## E   Transferability Performance

In this section, we test the transferability of our recovered reward machines when deployed on a different environment with the same labeling function. In particular, the new environment is similar to the Tabular GridWorld MDP studied in Section 5.1, except that the room assignment is changed, as shown in Figure 12a below. The dynamics are kept the same. In order to test transferability, we first generate a trajectory dataset using the ground-truth optimal policy in the new environment. Then, for each reward machine model, given by a varied $u_{\max}$, we construct a product policy according to Algorithm 4 (using the prefix tree policy of the original environment) and solve for a reward function that will be optimized in the new environment. We then compute the log-likelihood of the trajectory dataset given the corresponding optimized policies. The results are shown in Figure 12b, where (TRUE) is the log-likelihood of the ground-truth optimal policy in the new environment. It can be seen that the performance of the learned reward machine with 4 nodes almost matches that of the ground-truth reward machine. The results also display a graceful degradation in performance with decreasing number of nodes.

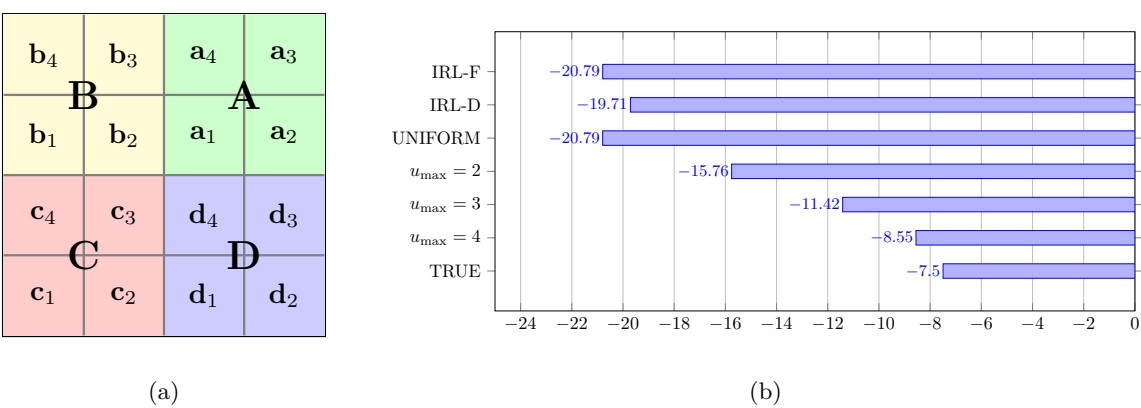

(a)                          (b)

Figure 12: (a): Modified labeled MDP. (b): Average log-likelihood of different models over 100 trajectories generated by the optimal policy in the new labeled MDP.

