# OpenReview forum: "Learning Reward Machines from Partially Observed Policies"
_TMLR — Accepted by TMLR_

### Review · Reviewer_EAZD · 2025-07-14

**Summary Of Contributions:**

The paper addresses the open problem of recovering an unknown reward machine (RM) purely from observations of an optimal policy in a labeled MDP, without ever seeing numeric rewards or RM states.
It introduces the prefix-tree policy (PTP) abstraction, which records the action distribution for every state * proposition-prefix pair observed from the expert; this is the only information available to the learner in their “partially observed” setting.

**Audience:**

Yes

**Claims And Evidence:**

Yes

**Requested Changes:**

### **Requested Changes**

* **Include quantitative baselines.**
  Add head-to-head comparisons with representative automata-learning (e.g., DFA inference) and inverse-RL algorithms (Bayesian IRL, AIRL, etc.) on at least one shared benchmark, reporting accuracy, sample efficiency, and runtime. Direct evidence of practical advantage is essential to justify acceptance.

* **Relax or test the exact-optimality assumption (Critical).**
  Either extend the theory to $\varepsilon$-optimal / noisy experts or demonstrate empirically how performance degrades with sub-optimal demonstrations. Without this, the guarantees rest on an unrealistic premise for real-world data.

* **Release anonymised code and data at review time (Critical).**
  Providing the promised Python + Z3 implementation, solver settings, and datasets now would let reviewers (and readers) verify claims and reproducibility.

* **Report sensitivity to the upper-node bound $u_{\max}$ (Strengthening).**
  Show how synthesis quality and runtime vary when $u_{\max}$ is misspecified, and propose an adaptive or heuristic selection rule (e.g., MDL, BIC, incremental growth).

* **Analyse the worst-case depth bound $l^{\star}=|S|u_{\max}^{2}$ in practice (Strengthening).**
  Offer tighter bounds for common RM structures or empirical depth-versus-error curves to clarify scalability on larger tasks.

**Strengths And Weaknesses:**

### **Strengths**

* **Solid Theory:** It proves an identifiability bound: a prefix-tree policy (PTP) observed to depth
  \$l^{\star}=|S|,u\_{\max}^{2}\$
  suffices to reconstruct the ground-truth RM up to policy-equivalence.

* **Robustness to Demonstration Noise:** Treating negative-example clauses as *soft* weights in a MAX-SAT variant allows correct recovery even when some demonstrations are mislabeled.

* **Diverse & Scalable Experiments:** Across four domains—grid-patrol, block-stacking, a continuous 2-D Reacher arm (1 M trajectories), and 200 real mouse-maze runs—the method attains exact or near-exact recovery with shallow depths and sub-second solver times.

---

### **Weaknesses / Points for Improvement**

* **Strong Optimality Assumption:** All guarantees rely on having the exact optimal policy; real-world demonstrations are often sub-optimal or noisy, weakening theoretical assurances.

* **Potentially Large Depth Bound:** The worst-case requirement \$l^{\star}=|S|u\_{\max}^{2}\$ scales quadratically with RM size and linearly with the MDP’s state count—prohibitive for complex tasks even if practice needs less.

* **Need for an Upper Bound \$u\_{\max}\$:** Both the theory and Algorithm 1 assume a tight prior on the number of RM nodes; guidance on selecting or adapting this bound is missing.

* **Scalability of the SAT Core:** The negative-example set can explode into hundreds of thousands of clauses, leading to many SAT calls; further heuristics or incremental solving strategies are only briefly mentioned.

---

> ### Author Response · Authors · 2025-08-18
>
> We extremely thank the reviewer for their valuable comments and suggestions. We have tried to address the weaknesses mentioned by the reviewer in our revised version. Below is our detailed response to each of the reviewer’s raised concerns.
>
> **Include quantitative baselines:**
>
>  In the updated version of the paper, we included a Max Causal Entropy IRL [12] baseline for the patrol task, which can be found in Table 6 of Section D in the Appendix. The input demonstration dataset is generated using an expert patrolling the rooms in the order A->B->C->D by following the ground-truth reward machine. While MaxEnt IRL is at the core of many SOTA IRL algorithms, the performance of this baseline is not of big surprise, as any standard IRL method will not be suitable for learning a reward function capable of solving this task. Concerning the “representative automata-learning (e.g, DFA inference)” baseline, we note that our learning setting is sufficiently different from most of the surveyed similar works, making the implementation of any representative baseline highly non-trivial. Below we present a survey of the related works and how they differ from our setting.
>
> References [1–8] generally assume knowledge of the reward signal, for which the problem becomes different variants of standard automata learning algorithms consistent with I/O traces. We do not assume the reward is observed.
>
> [1] Furelos-Blanco, Daniel, et al. "Induction and exploitation of subgoal automata for reinforcement learning." Journal of Artificial Intelligence Research 70 (2021): 1031-1116.
>
> [2] Furelos-Blanco, Daniel, et al. "Induction of subgoal automata for reinforcement learning." Proceedings of the AAAI Conference on Artificial Intelligence. Vol. 34. No. 04. 2020.
>
> [3] Learning and Exploiting Reward Machines for Reinforcement Learning.
>
> [4] Neider, Daniel, et al. "Advice-guided reinforcement learning in a non-Markovian environment." Proceedings of the AAAI Conference on Artificial Intelligence. Vol. 35. No. 10. 2021.
>
> [5] Xu, Zhe, et al. "Joint inference of reward machines and policies for reinforcement learning." Proceedings of the International Conference on Automated Planning and Scheduling. Vol. 30. 2020
>
> [6] Xu, Zhe, et al. "Active finite reward automaton inference and reinforcement learning using queries and counterexamples." Machine Learning and Knowledge Extraction: 5th IFIP TC 5, TC 12, WG 8.4, WG 8.9, WG 12.9 International Cross-Domain Conference, CD-MAKE 2021, Virtual Event, August 17–20, 2021, Proceedings 5. Springer International Publishing, 2021.
>
> [7] Hasanbeig, Mohammadhosein, et al. "Deepsynth: Automata synthesis for automatic task segmentation in deep reinforcement learning." Proceedings of the AAAI Conference on Artificial Intelligence. Vol. 35. No. 9. 2021.
>
> [8] Toro Icarte, Rodrigo, et al. "Learning reward machines for partially observable reinforcement learning." Advances in neural information processing systems 32 (2019).
>
> References [9–10]  do not assume the reward signal is known but are constrained to Markovian tasks. All of our tasks are Non-Markovian.
>
> [9] Camacho, Alberto, et al. "Reward machines for vision-based robotic manipulation." 2021 IEEE International Conference on Robotics and Automation (ICRA). IEEE, 2021.
>
> [10] Baert, Mattijs, Sam Leroux, and Pieter Simoens. "Reward Machine Inference for Robotic Manipulation." arXiv preprint arXiv:2412.10096 (2024).
>
> Reference [11] assumes that the reward machine state (or automata state) is observed in the training set, which we do not assume.
>
> [11] Araki, Brandon, et al. "Learning to plan with logical automata." (2019).
> [12] Ziebart, Brian D. Modeling purposeful adaptive behavior with the principle of maximum causal entropy. Carnegie Mellon University, 2010.
>
> We summarize this body of literature in the table below.
>
> | Setting                         | Reward not observed | Reward machine state not observed | Non-Markovian Tasks |
> |---------------------------------|---------------------|----------------------------------|---------------------|
> | [1–8]                           | ❌                  | ✔                                | ✔                   |
> | [9,10]                          | ✔                  | ✔                                | ❌                  |
> | [11]                            | ✔                  | ❌                                | ✔                   |
> | Ours                            | ✔                  | ✔                                | ✔                   |

---

> > ### Author Response · Authors · 2025-08-18
> >
> > The major differences mentioned above make it hard to directly compare the advantage of our method over others as we believe the problem we are solving is novel in nature.
> >
> > **Relax or Test the exact-optimality assumption:**
> >
> > This is a good point that the reviewer raises. While we do mention at multiple places in the sections building up to the algorithm that we require prefix tree policies generated from the optimal policy, the optimality of the policy is not necessary for most of our results or theorems to hold. We emphasize this in Remark 2 as well. To further illustrate this point, we added a new example to Section C of the appendix. In particular, given the two-node reward machine considered there, we generated a completely random policy at each node, which challenges the optimality assumption. Still, the framework recovers the ground truth reward machine model up to renaming. Another example is Section 5.4, where we applied the framework to a real world mice dataset, where again no policy optimality is assumed.
> >
> > **Release Anonymised code:**
> >
> > the code is available in the supplementary materials now.
> >
> > **Report Sensitivity to upper bound:**
> >
> > We added Appendix D to tackle this point. In particular, we showed how the upper bound specified affects the synthesis quality by looking at the performance of policies resulting from misspecifications. Overall we observed a degradation in quantitative performance measured by the average rewards accumulated in a rollout, while qualitatively the learned reward machine models still capture a good representation of the task.
> >
> > **Analyse the worst-case depth bound:**
> >
> > Obtaining a tighter worst-case depth bound seems elusive at this time but will be considered in our future work. We indicated this limitation clearly in the newly introduced limitations and future work section of our paper. It is worth mentioning that this bound doesn’t take non-stuttering into account, so a simple reduction of the bound is taking modulo repeated APs.

---

### Review · Reviewer_zSDg · 2025-08-08

**Summary Of Contributions:**

This paper addresses a problem in the field of Inverse Reinforcement Learning (IRL): learning a structured reward function, represented as a Reward Machine (RM), from an expert's policy without access to direct reward signals or the internal states of the RM. The core contribution is a framework that can infer the structure of the underlying RM by only observing the expert's actions and the resulting sequences of high-level "atomic propositions" from the environment.

The  methodology involves three key steps:
   - **Prefix Tree Policy** (PTP): A representation that maps a history of atomic proposition sequences (a prefix) and a current state to a distribution over actions. This captures how an expert's policy changes based on task progress.
   -  **Equivalence and SAT Formulation**: They formalise the problem by defining a policy-equivalence class for RMs, as multiple RMs can produce the same observable policy. The search for a consistent RM within this class is then encoded as a Boolean Satisfiability (SAT) problem. Constraints for the SAT solver are derived from the structure of RMs and, crucially, from "negative examples", namely pairs of proposition sequences that lead to different observed policies, implying they must terminate at different nodes in the true RM.
   - **Learning from Demonstrations**: The framework is extended to the more practical setting of learning from a finite set of expert demonstrations. To handle the noise and uncertainty inherent in policy estimates from finite data, the SAT problem is relaxed into a weighted MAX-SAT problem, providing robustness against incorrectly identified negative examples.


  The authors provide a theoretical proof that their algorithm can recover a policy-equivalent RM given a policy known up to
  a sufficient, finite depth. They validate their approach on a diverse set of experiments, including discrete grid worlds, a continuous robotic arm simulation, and a real-world dataset of mouse navigation, demonstrating its effectiveness and generality.

**Audience:**

Yes

**Claims And Evidence:**

Yes

**Requested Changes:**

**Low-Level**
- (major) I would strongly suggest to provide references and background for SAT and (minor) to frame sub subsections as paragraphs, for readability.
- Figure 1(c) is kind of far from its actual reference, I would suggest to adjust for this.

**High-Level**:
- I would strongly suggest to add in the introduction a high level description with examples of the setting of interest, naming some examples of what solving for such task might be, for instance by describing the experimental problems at hand.
- It would be beneficial to add a more explicit discussion on the scalability limitations of the
      approach.
- Some guidance on setting the key. hyperparameter α (the confidence threshold for negative examples) would be valuable. A brief discussion on the sensitivity  of the results to this parameter would improve the paper's practical utility.
- The paper mentions using weighted MAX-SAT to handle false positives but I did not find details on how the weights are assigned. A brief sentence explaining that  more explicitly would be nice.
- A dedicated section on "Limitations and Future Work" would be a valuable addition.

**Strengths And Weaknesses:**

**Strenghts**:

   * **Novel and Important Problem Formulation**: The paper tackles a well-motivated problem. Learning structured
     rewards from policy observations alone is a significant step beyond traditional IRL methods that rely on reward
     observations or full state-space trajectories. This has interesting implications for autonomous systems and behavior analysis.
   * **Theoretical Rigor**: The work is theoretically sound. The formalization of the Prefix Tree Policy and the characterization
     of the RM equivalence class provide a solid foundation for the problem. The main theoretical result (Thm 1), which
     guarantees the recovery of an equivalent RM, is a powerful and relevant statement.
   * **Nice Algorithmic Approach**: The use of a SAT-based formulation is a nice tweak. It turns a complex
     search problem into a well-understood constraint satisfaction problem, for which highly optimized solvers exist. Encoding
     policy differences as "negative example" constraints is a particularly insightful way to prune the search space.
   * **Robustness and Practicality**: The extension to learning from finite demonstrations via weighted MAX-SAT is crucial for
     real-world applicability.
   * **Satisfactory Empirical Validation**: By testing the framework on discrete, continuous,
     and real-world biological data, the authors effectively demonstrate the versatility and robustness of their approach. The
     successful application to the continuous Reacher arm and the real-world mouse data is particularly nice to me.

**Weaknesses**:

   * **Scalability Concerns**: The primary weakness, common to many SAT-based approaches, is scalability. The theoretical bound for
     the required policy depth ($l > |S| u^2_{max}$) can be very large for complex environments. Furthermore, the number of
     potential negative examples can grow exponentially with this depth, potentially making the SAT instance intractable for
     problems with large state spaces or highly complex RMs.
   * **Dependence on $u_{max}$**: The algorithm requires an upper bound on the number of RM nodes as an input. While
     Algorithm 1 iterates the number of nodes n upwards, the theoretical guarantees are framed in terms of a known u_max. In a
     real-world setting where the true RM is unknown, choosing an appropriate u_max (or knowing when to stop the iteration of
     n) is a non-trivial challenge. An insufficient u_max could lead to the recovery of a non-equivalent, overly simplistic RM.

   * **Assumption of Deterministic RMs**: The framework assumes that the underlying task is representable by a deterministic RM.
     While this is a powerful and common model, some complex, stochastic, or history-dependent tasks might not be perfectly
     captured by this structure.

---

> ### Author Response · Authors · 2025-08-18
>
> We sincerely thank the reviewer for the careful reading and the many thoughtful suggestions. We’re grateful that you found the problem formulation novel and important, the theory sound, the SAT approach insightful, and the empirical results compelling. We have revised the paper accordingly. Below we describe how we addressed each of the requested changes.
>
> **Low-level**
>
> ***Background on SAT***
> Thank you for the suggestion. Because SAT is a very well-established topic, we opted to keep the background succinct to conserve space. We now include a brief, contextual sentence and a citation to a standard reference in section 3.1.
>
> ***Sub-subsections → paragraphs***
> Done. Thank you for the suggestion.
>
> ***Figure 1(c) placement***
> We do agree that the placement of figure 1.c closer to its actual reference makes the reading easier but we opted to keep it in its current location in order to preserve the coherence of the three-panel narrative.
>
> **High-level**
>
> ***High-level description with examples (Introduction)***
> Done. We expanded the second paragraph of the Introduction, with concrete examples that we later build on in the main body or experiments section of the paper to further emphasize the motivation for learning RMs directly from observed policy behavior.
>
> ***Scalability discussion***
>  Added. We created a new Limitations and Future Work section explicitly discussing: (i) how required policy depth can expand the constraint set, (ii) growth in potential negative examples, and (iii) practical implications for SAT instance size.
>
> ***MAX-SAT weights***
> Clarified. We now state explicitly in Section 4.2 that the weighted MAX-SAT uses uniform weights for all soft (negative-example) constraints in our experiments.
>
> ***Sensitivity to α (confidence threshold)***
> Added. Table 3 reports a sensitivity analysis over α. Across tasks, we observe that α in the range 0.01 to 0.00001 yields very similar performance, with negligible differences in reconstruction accuracy and policy likelihood.

---

### Review · Reviewer_Rfpt · 2025-08-09

**Summary Of Contributions:**

This paper studies how to learn reward machines when you cannot directly observe the rewards or the machine’s internal states, but only see part of the optimal policy. The authors introduce the idea of a prefix tree policy, which records what the agent would do in a given MDP state after seeing a certain sequence of atomic propositions. They define what it means for two reward machines to be equivalent in terms of the policies they produce, and they show how to recover a minimal equivalent machine. The main technical tool is a SAT formulation that searches for a machine structure consistent with the observed policy. They prove that if you have a long enough prefix tree policy, the method can recover the exact equivalent machine. They also extend the approach to work from demonstration data by filtering likely errors and using a MAX-SAT variant. Experiments on grid worlds, block worlds, 2D reacher, and a real mouse navigation dataset show the method can find the correct machine up to renaming.

**Audience:**

Yes

**Broader Impact Concerns:**

No ethical concerns

**Claims And Evidence:**

Yes

**Requested Changes:**

1. The paper is difficult to follow, especially for audience without much background of reward machines. The authors should make the writing clearer, motivate the formulation of the problem first, and add intuitive motivations/explanations/examples alongside the formal descriptions.
2. The method assumes that the atomic propositions and labeling are completely correct. The authors should discuss how robust the approach is to incorrect or noisy labels and, if possible, test it on data with imperfect labeling.
3. The experiments mostly test recovery in known settings. The authors should also show results in cases where the true reward machine is unknown or where the learned machine is applied to a different environment with the same propositions.
4. The mouse navigation experiment is only analyzed qualitatively. The authors should add some form of quantitative evaluation for this case, even if it is indirect, to strengthen the real-world evidence.
5. Relevant baselines, including standard IRL methods and Behavioral Cloning, should be discussed in more depth and compared against in the experiments to facilitate understanding the impact of this work.

1-4 is important for my recommendation, 5 is nice to have.

**Strengths And Weaknesses:**

Strengths
- The paper is the first to address the problem of learning reward machines when only a partially observed policy is available, filling a gap in prior work that assumed access to rewards or full policy information.
- It provides formal guarantees that, under specific assumptions, the method will recover a reward machine equivalent to the ground truth, giving the approach a solid theoretical foundation.
- The work clearly separates the process of learning the machine’s structure from the process of estimating the numerical reward parameters, which makes the method modular and easier to adapt.
- In practice, the approach is made more robust by using a MAX-SAT formulation that can tolerate noisy or imperfect data in the demonstration setting.

Weaknesses
- The method assumes that the set of atomic propositions and the labeling function mapping MDP states to these propositions are completely known and correct, which may be unrealistic in many applications.
- It requires the reward machine to be deterministic and to have a defined transition for every possible proposition in every state, which are strong assumptions.
- The approach depends on the observed policy being generated by a maximum-entropy optimal policy; deviations from this assumption could reduce identifiability and lead to incorrect machines.
- The theoretical bound on the prefix length needed for guaranteed recovery is very large, making it impractical for large problems, and the paper does not provide a practical way to decide when enough data has been collected.
- The noise handling in the demonstration setting relies on heuristic filtering and weighted MAX-SAT, without formal guarantees on accuracy in the presence of limited or noisy data.
- The paper’s writing and presentation are not very clear, the lengthy definitions with dense symbols are hard to follow without explanation of the intuition or implication of them
- Most experiments are verification-style, where the authors design environments with a known reward machine, generate data from it, and check whether their method recovers it; this mainly tests correctness rather than usefulness in real-world or transfer scenarios.
- The only real-world experiment, using mouse navigation data, is interpreted qualitatively rather than validated quantitatively, so it does not fully demonstrate the method’s effectiveness when the ground truth is unknown.
- The paper does not discuss related work in imitation learning or inverse reinforcement learning in sufficient depth, nor does it compare the proposed approach against these baselines. While the focus here is on recovering reward machine structure, it would be valuable to situate the method more clearly within the broader IL and IRL literature and to highlight its advantages over existing approaches.

---

> ### Author Response · Authors · 2025-08-18
>
> We thank the reviewer for their detailed and constructive feedback, and for recognizing the novelty, theoretical grounding, and modularity of our approach. We appreciate the specific suggestions, which have helped us identify several points for clarification and improvement.
>
> Among the listed weaknesses we would like to clarify one. We respectfully disagree with the comment on determinism and input completeness of the reward machine being a strong assumption. We believe this assumption is very natural to have a well-defined problem. Please note that if the reward machine is non-deterministic or blocking, an infinite horizon RL problem with such an RM is not well-defined. Some ways to make the RL problem well-posed for RMs that do not satisfy these two assumptions may include determinizing the RMs (using, e.g., the equivalence of deterministic and non-deterministic finite state automata) or adding a sink node to a blocking RM, which essentially amount to creating modified RMs consistent with our assumptions.
> We tried to address the other weaknesses following the suggestions in the required changes.
>
> As for the requested changes:
>
> $1) \textbf{ Clarity and Motivation:}$ the second paragraph of the introduction aims at describing the motivation behind reward machine learning, as well as why reward machines are needed for temporally extended (non-Markovian) settings. We expanded this paragraph to further clarify the motivation by using examples. We also note that the last paragraph of Section 2.1 demonstrates many of the key concepts and definitions, using a running example, which we believe will aid the readers not familiar with the reward machine formalism.
>
>
> $2) \textbf{ Assumption of Correct Labeling:}$ We acknowledge that our current framework assumes access to the ground-truth atomic propositions and labeling function, as is standard in much of the literature. This limitation is now explicitly stated in a dedicated Limitations and Future Work section and we state that handling unknown or noisy labeling is left for future work.
>
> $\textbf{ 3) Experiments Beyond Recovery in Known Settings:}$ We note that two of our existing domains, the Reacher environment and the Labyrinth dataset, involve unknown ground-truth reward machines, thus addressing part of this concern. That is, the data for these examples are not generated by solving a forward RL problem with a reward machine. Hence we believe these settings go beyond pure recovery experiments. Furthermore, we followed the reviewers suggestion to add examples where the learned machine is applied to a different environment with the same propositions. This experiment on transferability to new environments is added in Appendix E.
>
> $\textbf{4) Quantitative Analysis of Mouse Navigation:}$ Thank you for this suggestion. We extended the mouse navigation evaluation with a quantitative measure following the reviewer’s request. In particular, we computed the likelihood of held-out trajectories under policies learned from our recovered reward machine, and compared this against policies learned using a standard IRL baseline.
>
> $\textbf{5) IRL baselines:}$ We included IRL baselines in several examples further demonstrating how capturing the non-Markovian structure via a reward machine improves upon standard IRL baselines. We believe behavior-cloning is a bit tangential to our motivation. Given that it is hard to do justice to large literature in that domain, we opted for not including a comparison with BC baselines.

---

### Review · Reviewer_Q4TZ · 2025-08-09

**Summary Of Contributions:**

At a high level, this paper identifies a sufficient condition toward inverse RL: the problem of learning the reward function given the optimal policy (where we only observe the atomic proposition associated with the reward function), in the setting where the reward function is given by a reward machine whose transitions depend on certain prefix trees.

**Audience:**

Yes

**Broader Impact Concerns:**

-

**Claims And Evidence:**

Yes

**Requested Changes:**

-

**Strengths And Weaknesses:**

Strengths:
1. The paper is very well written, and the reward machine setting is clearly defined.
2.  Rigorous proofs are given for Theorem 1 which justify the equivalence between the learned policy and the policy of the reward machine
3. This paper strictly improves on previous work, so it is a generally important/helpful line of research.

Weaknesses:

1. The setting cannot handle (and does not address) vector rewards
2. The setting only considers deterministic reward machines

---

> ### Author Response · Authors · 2025-08-18
>
> We thank the reviewer for their supportive and constructive feedback, and for recognizing both the clarity of our presentation and the significance of the line of research.
>
> Regarding the concern about vector-valued rewards: indeed, our framework can handle arbitrary reward functions on each node of the reward machine. As noted in Remark 1 of the paper, the two-step nature of our approach means that the reward machine model learning step remains unchanged, and only the reward learning step needs to be adapted. Moreover, we note that vector-valued rewards inherently introduce more ambiguity in inverse reinforcement learning, as discussed in [1,2,3].
>
> [1]: Cao, Haoyang, Samuel Cohen, and Lukasz Szpruch. "Identifiability in inverse reinforcement learning." Advances in Neural Information Processing Systems 34 (2021): 12362-12373.
> [2]:Rolland, P., Viano, L., Schürhoff, N., Nikolov, B., & Cevher, V. (2022). Identifiability and generalizability from multiple experts in inverse reinforcement learning. Advances in Neural Information Processing Systems, 35, 550-564.
> [3]: Shehab, M. L., Aspeel, A., Arechiga, N., Best, A., & Ozay, N. (2024, June). Learning true objectives: Linear algebraic characterizations of identifiability in inverse reinforcement learning. In 6th Annual Learning for Dynamics & Control Conference (pp. 1266-1277). PMLR.

---

### Decision · Action_Editor_oBtE · 2025-10-06

**Recommendation:** Accept as is

**Additional Comments:**

The authors have addressed all reviewers' concerns.

**Audience:**

Yes

**Audience Explanation:**

All reviewers agree that the manuscript has an audience. More formalisms and theoretical justification together with experimental evidence are definitely relevant to the inverse reinforcement learning subcommunity and the TMLR readership.

**Claims And Evidence:**

Yes

**Claims Explanation:**

All reviewers agree that the manuscript presents appropriate evidence for its claims. On inverse reinforcement learning, the manuscript proposes to identify the true reward machine from partially observed optimal policies directly without the need to observe the rewards or the machine's state. Formalisms such as prefix tree policies, policy equivalence and a SAT-based encoding are used. Theoretical justification as well as experimental evidence is provided.